# SH3RF3 promotes breast cancer stem-like properties via JNK activation and PTX3 upregulation

Peiyuan Zhang[1,2,6], Yingjie Liu[1,2,6], Cheng Lian[1], Xuan Cao[3], Yuan Wang[1], Xiaoxun Li[1], Min Cong[1], Pu Tian[1], Xue Zhang[1], Gang Wei [3], Tong Liu[4,5✉] & Guohong Hu [1✉]

Cancer stem-like cells (CSCs) are the tumorigenic cell subpopulation and contribute to cancer recurrence and metastasis. However, the understanding of CSC regulatory mechanisms remains incomplete. By transcriptomic analysis, we identify a scaffold protein SH3RF3 (also named POSH2) that is upregulated in CSCs of breast cancer clinical tumors and cancer cell lines, and enhances the CSC properties of breast cancer cells. Mechanically, SH3RF3 interacts with the c-Jun N-terminal kinase (JNK) in a JNK-interacting protein (JIP)-dependent manner, leading to enhanced phosphorylation of JNK and activation of the JNK-JUN pathway. Further the JNK-JUN signaling expands CSC subpopulation by transcriptionally activating the expression of Pentraxin 3 (PTX3). The functional role of SH3RF3 in CSCs is validated with patient-derived organoid culture, and supported by clinical cohort analyses. In conclusion, our work elucidates the role and molecular mechanism of SH3RF3 in CSCs of breast cancer, and might provide opportunities for CSC-targeting therapy.

[1] Shanghai Institute of Nutrition and Health, Shanghai Jiao Tong University School of Medicine (SJTUSM) & Chinese Academy of Sciences, Shanghai, China. [2] CAS Key Laboratory of Tissue Microenvironment and Tumor, Shanghai Institute of Nutrition and Health, University of Chinese Academy of Sciences, Chinese Academy of Sciences, Shanghai, China. [3] CAS Key Laboratory of Computational Biology, Collaborative Innovation Center for Genetics and Developmental Biology, CAS-MPG Partner Institute for Computational Biology, Shanghai Institute of Nutrition and Health, Chinese Academy of Sciences, Shanghai, China. [4] Department of Breast Surgery, Harbin Medical University Cancer Hospital, Harbin, China. [5] The Institute of Cancer Prevention and Treatment, Harbin Medical University, Harbin, China. [6] These authors contributed equally: Peiyuan Zhang, Yingjie Liu. ✉email: liutong@hrbmu.edu.cn; ghhu@sibs.ac.cn

Cancer represents a major threat of human health. Due to high tumor heterogeneity, the efficacy of most cancer therapies is often hindered by the existence of minor subpopulation of cancer cells with enhanced therapy resistance, which leads to tumor recurrence and eventually therapy failure. The CSC theory offers one explanation of tumor heterogeneity[1,2] and states that CSCs, also referred as tumor-initiating cells, are the cell subpopulation in tumors equipped with tumor-initiating capabilities of unlimited self-renewal and differentiation[3,4]. More importantly, CSCs display enhanced resistance to chemotherapies and radiotherapies, and are often considered to account for poor therapeutic response, recurrence, and metastasis. The CSC theory has been well tested in breast cancer. Identified early in this century, breast cancer stem-like cells (BCSCs) were originally characterized by high expression of CD44 and low or negative expression of CD24[5]. Later, another type of BCSCs was also discovered with high aldehyde dehydrogenase (ALDH) activity[6]. Although a number of studies have identified the intracellular and microenvironmental cues that regulate breast cancer stemness[7,8], our understanding toward CSCs in breast cancer remains incomplete. Elucidating the molecular underpinning of BCSCs, especially the common regulators of different BCSC types with clinical relevance, will be important for rational designing of CSC-targeting strategies in cancer treatment.

SH3RF3 (SH3 domain containing ring finger 3), also known as POSH2 (plenty of SH3 domains protein 2), is a recently identified protein with four Src homology 3 (SH3) domains and a Ring finger domain[9]. SH3RF3 is found to function as a scaffold interacting with PAK2 and RAC1 through its SH3 domains[9,10]. In addition, the Ring finger domain might also confer a self-ubiquitinating enzymatic activity to the protein[10]. Clinical analyses have shown that SH3RF3 expression is associated with acute lymphoblastic leukemia[11], HIV-associated neurocognitive disorder[12], and late-onset familial Alzheimer's disease[13]. However, the functional role and mechanism of SH3RF3 in pathological processes, especially in cancer, are largely unexplored. In this study, we found SH3RF3 could upregulate the expression of Pentraxin 3 (PTX3) to promote stem-like traits of breast cancer cells. PTX3 is a secretory protein belonging to the pentraxin family. As a pattern recognition receptor, PTX3 plays critical roles in innate immunity[14], tissue repair, and remodeling[15]. PTX3 dysregulation is also widely observed in cancer[16–20]. Many studies have established the potential of PTX3, especially its serological level, as a biomarker for a number of pathological conditions including cancer[16,17,20,21]. However, the roles of PTX3 in various cancers have been seemingly controversial, with reports showing both tumor-suppressing and promoting effects of PTX3 in tumor stemness, growth and metastasis[22]. Therefore, a clear definition of its functions in specific cancer types is of practical importance for clinical application of PTX3 in diagnosis and therapeutics.

In this study, we show that the scaffold protein SH3RF3 is highly expressed in breast cancer CSCs and reveal its function to enhance the cancer cell stemness by regulating PTX3.

## Results

**SH3RF3 is associated with CSC properties in breast cancer.** To identify regulatory molecules in BCSCs, we sorted HMLER, a human breast cancer cell line often used in CSC studies[23], into a CD44+CD24− (termed as CD44H subsequently) subpopulation and a CD44−CD24+ (termed as CD44L subsequently) subpopulation via flow cytometry (Fig. 1a). The CD44L cells were further cultured in consecutive tumorsphere passages, resulting in a new subpopulation named as CD44LS. Both the CD44H and CD44LS sublines displayed enhanced tumorsphere formation (Fig. 1b) and resistance to chemotherapeutic drugs including paclitaxel and doxorubicin (Fig. 1c), corroborating CSC enrichment in these two sublines. Interestingly, CD44H and CD44LS showed distinct expression pattern of CSC markers. While CD44H was mainly CD44+CD24− and contained few ALDH+ cells, the CD44LS cells displayed much higher expression of ALDH (Fig. 1d). Notably, although sphere culturing of CD44L cells led to gradual increase of CD44 expression (Supplementary Fig. 1a), the CD44+CD24− content of CD44LS remains much lower than that of CD44H cells (Fig. 1d). These observations were in line with the previous notion that CD44+CD24− and ALDH+ mark distinct BCSC subpopulations[24]. Then we analyzed the transcriptomic profiles of CD44L, CD44H, and CD44LS via RNA sequencing, and identified 588 upregulated and 706 down-regulated genes in CD44H and CD44LS as compared with CD44L (Supplementary Data File 1).

To ensure the clinical relevance of these CSC-associated genes, we further analyzed the transcriptomic datasets of two clinical cohorts. First we analyzed the Cancer Genome Atlas (TCGA) breast cancer cohort by single-sample gene set enrichment analysis (ssGSEA[25]), with a CD44+CD24− BCSC signature previously identified in human breast carcinomas[26], and then the genes with expression highly correlated with the BCSC ssGSEA scores in the TCGA samples were selected. In addition, we analyzed the expression profiles of a cohort of patient-derived tumorspheres versus the corresponding primary tumors (GSE7515[27]), to identify the genes differentially expressed in the tumorspheres. These clinical analyses resulted in 76 genes (41 upregulated and 35 downregulated) that were associated with both the CD44+CD24− signature and the tumorsphere-forming capacity in clinical tumors (Supplementary Table 1).

Finally we examined the overlap of these 76 genes with the above CSC-related genes identified in HMLER sublines, and found 21 genes (12 upregulated and 9 downregulated) in common (Fig. 1e). Many of the genes in the list, including GLI2[28], COL6A2[29], GRP124[30], SERPINF1[23,31], MMP2[32], TIMP2[33], GRHL2[34], and ESRP1[35], have been previously shown to regulate stemness of normal or cancer cells, or epithelial–mesenchymal transition (EMT), a process highly related to CSC properties[36]. Notably, SH3RF3, which had not been previously implicated in CSC regulation, was among the genes highly upregulated in CSC-enriched tumors and cancer cell sublines (Fig. 1e). Further real-time reverse transcription PCR (qRT-PCR) and Western blot analyses confirmed the elevated expression of SH3RF3 in CD44H and CD44LS sublines of HMLER (Fig. 1f). In addition, when we isolated CD44+CD24− and CD44−CD24+ subpopulations of two additional human cell lines HMLE and MCF10AT, upregulation of SH3RF3 in the CD44+CD24− subpopulation was also observed (Fig. 1f), indicating a role of SH3RF3 in BCSC regulation.

**SH3RF3 promotes CSC properties of breast cancer cells.** To investigate the functional role of SH3RF3 in BCSCs, SH3RF3 was overexpressed in HMLE, MCF10AT and MDA-MB-231 cells (Fig. 2a), followed by flow cytometry analyses of CSC contents in these cell lines. SH3RF3 overexpression led to CD44+CD24− cell expansion in HMLE and MCF10AT. The ALDH+ subpopulation in MDA-MB-231 was also increased (Fig. 2b). In addition, SH3RF3 overexpression significantly enhanced tumorsphere formation in all three cell lines (Fig. 2c). In Py8119, a murine breast cancer cell line derived from the PyMT-driven tumors, Sh3rf3 upregulation was also observed in the ALDH+ subpopulation versus the ALDH− counterpart (Supplementary Fig. 1b), and Sh3rf3 overexpression increased the capacity of tumorsphere formation by Py8119 (Supplementary Fig. 1c and d). Further, we

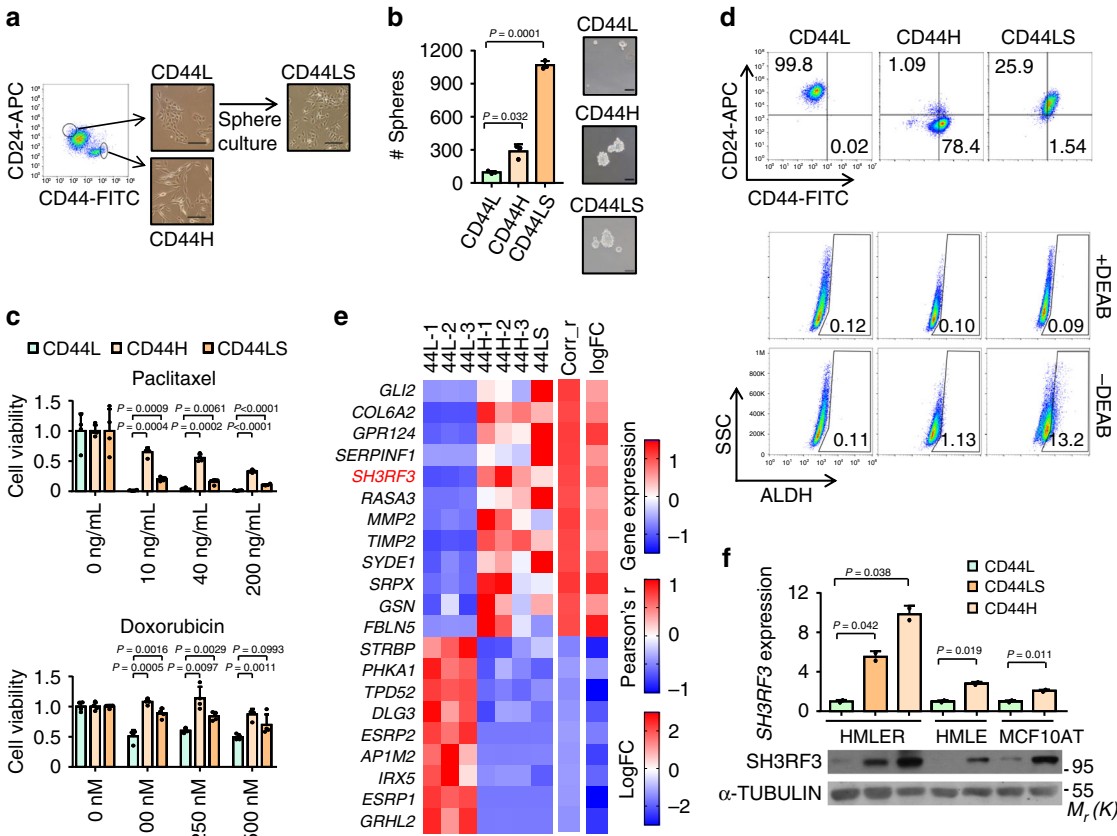

**Fig. 1 SH3RF3 expression is associated with BCSC properties. a** Flow chart and representative images of HMLER subline derivation. Scale bar, 100 μm. **b** Quantitation (*n* = 3 culturing experiments) and representative images of tumorspheres in HMLER sublines. Scale bar, 100 μm. **c** MTT assays of HMLER sublines after paclitaxel (0–200 ng/mL) and doxorubicin (0–500 nM) treatment in various concentrations (*n* = 4 treatment experiments). **d** Flow cytometry analyses of CD44$^+$CD24$^-$ and ALDH$^+$ CSCs in HMLER sublines. Numbers in the cytometry charts indicate the percentages of corresponding subpopulations. **e** Genes associated with CSC properties in clinical tumors and HMLER sublines. Heatmap showed their expression in HMLER sublines, Pearson's *r* correlation to CSC signature in TCGA tumors (Corr_r) and log-fold changes of expression in tumorspheres versus primary tumors (logFC). **f** SH3RF3 expression in different sublines derived from HMLER, HMLE and MCF10AT (*n* = 3 cell culturing). Data represent mean ± SD. Statistical significance was determined by two-tailed unpaired *t*-test (**b**, **c**, and **f**). The experiments in (**b**) and (**f**) were repeated three times independently with similar results, and the data of one representative experiment are shown. Source data are provided as a Source data file.

analyzed the in vivo tumorigenic capacity of *SH3RF3*-overexpressing cancer cells via limiting dilution assays. Various numbers of MDA-MB-231 control or *SH3RF3*-overexpressing cells were orthotopically transplanted into female immunodeficient NSG mice. When 1000 cells were injected, all mice developed tumors and no difference was observed between the two groups. However, when fewer cells were inoculated, a stronger tumor-initiating capability was observed in the overexpression group. While only 10% of the mice injected with 40 control cells succumbed to tumors at day 28, the tumorigenic rate was 60% for the overexpression group at the same time point, with the CSC frequency nearly tripled after *SH3RF3* overexpression (Fig. 2d). The tumor volumes were also obviously enhanced in mice inoculated with overexpressing cells (Fig. 2e). A similar phenomenon was also observed for HMLER cells in which *SH3RF3* overexpression led to enhance in vivo tumor incidence in the limiting dilution assay (Supplementary Fig. 1e and f). These results suggest that *SH3RF3* overexpression could facilitate CSC properties in breast cancer cells.

To further test whether SH3RF3 is essential for CSC maintenance in breast cancer, *SH3RF3* was knocked down in HMELR-CD44H and MCF10CA1h cell lines by multiple small interfering RNAs (siRNAs) or short hairpin RNAs (shRNAs) (Fig. 3a). Flow cytometry analyses showed an obvious shift to lower CD44 expression in HMELR-CD44H and a marked

decrease of ALDH$^+$ fraction in MCF10CA1h after *SH3RF3* knockdown (Fig. 3b). *SH3RF3* knockdown also impaired the capability of both HMELR-CD44H and MCF10CA1h to form tumorspheres (Fig. 3c). More importantly, limiting dilution assays showed that *SH3RF3* knockdown diminished the tumorigenic capability of MCF10CA1h cells in mice after orthotopic transplantation (Fig. 3d, e). Altogether, these results suggested the role of SH3RF3 in CSC promotion and maintenance in breast cancer cells.

**PTX3 is regulated by SH3RF3 and enhances BCSC properties.** To elucidate the mechanism of SH3RF3 in BCSC regulation, we analyzed the transcriptomic profiles of MCF10AT cells by RNA sequencing to identify the genes with expression significantly changed after *SH3RF3* overexpression. We further overlapped these genes with those differentially expressed in HMLER sublines. The analysis resulted in a list of 24 genes associated with both *SH3RF3* expression and BCSC properties (Fig. 4a). In the list, *PTX3*, encoding a secreted protein of the pentraxin family, was one of the top ranked genes. The regulation of *PTX3* by *SH3RF3* was verified in RNA and protein levels after *SH3RF3* overexpression and knockdown (Fig. 4b). In addition, analysis of the Cancer Cell Line Encyclopedia expression database[37] showed a strong positive correlation of *SH3RF3* and *PTX3* expression in

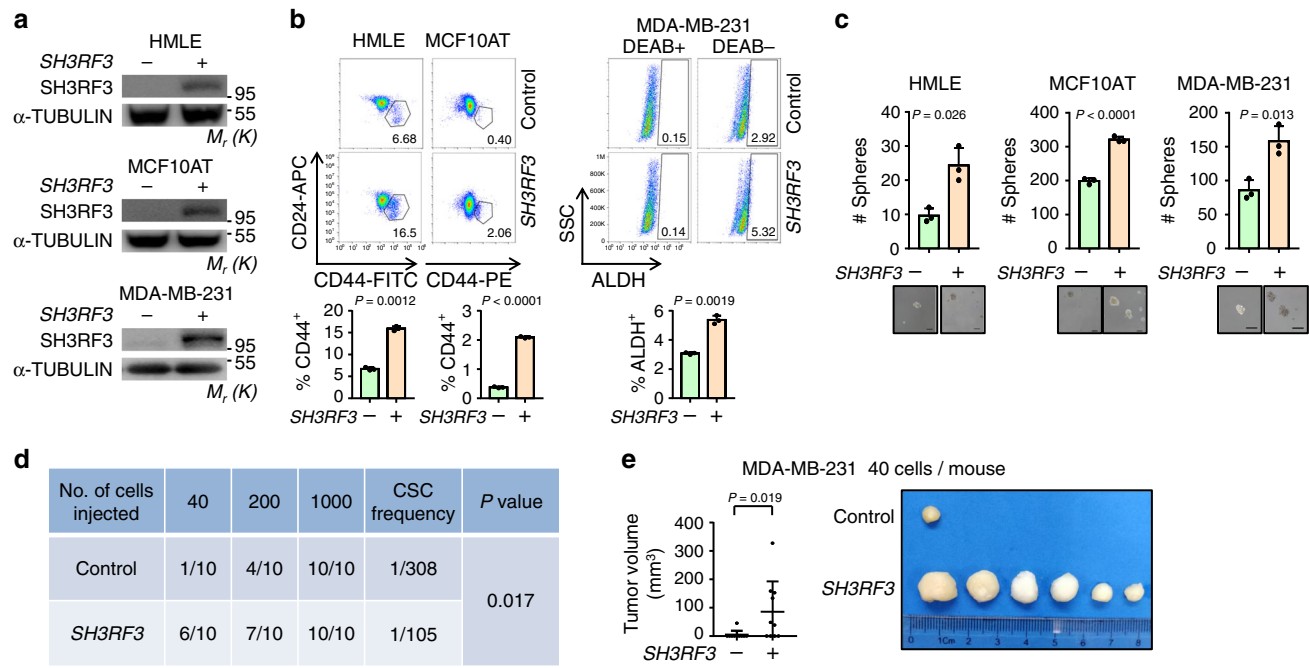

**Fig. 2 SH3RF3 promotes CSC properties of breast cancer cells. a** *SH3RF3* overexpression in HMLE, MCF10AT and MDA-MB-231 cells. **b** Flow cytometry analyses of the CD44+CD24− subpopulations in HMLE and MCF10AT, and the ALDH+ subpopulation in MDA-MB-231. Numbers in the flow cytometry charts indicate the CSC percentages (n = 3 culturing experiments). **c** Quantitation and representative images of tumorsphere formation in *SH3RF3*-overexpressing breast cancer cells (n = 3 culturing experiments). Scale bar,100 μm. **d** In vivo tumor formation of the mice injected with serial dilutions of MDA-MB-231 cells at day 28. **e** Tumor images and volumes (n = 10 mice) in mice injected with 40 control and *SH3RF3*-overexpressing MDA-MB-231 cells. Data represent mean ± SD. Statistical significance was determined by two-tailed unpaired *t*-test (**b** and **c**), chi-squared test (**d**) or Mann–Whitney U-test (**e**). The experiments in (**a**), (**b**) (upper), and (**c**) were repeated three times independently with similar results, and the data of one representative experiment are shown. Source data are provided as a Source data file.

breast cancer cell lines (Fig. 4c), further confirming that *PTX3* is a downstream gene regulated by *SH3RF3*.

Next, the functional role of PTX3 in BCSCs was studied. *PTX3* overexpression in HMLE, HMLER and MCF10AT (Fig. 4d and Supplementary Fig. 2a) led to obvious enhancement of tumorsphere formation by these cell lines (Fig. 4e and Supplementary Fig. 2b). *PTX3* overexpression resulted in increases of CD44+CD24− subpopulation of HMLE (Fig. 4f, g) and in vivo tumorigenesis by HMLER (Fig. 4h). Further, the transcriptomes of control and *PTX3*-overexpressing HMLER cells were profiled by RNA sequencing, followed by gene set enrichment analysis (GSEA) of the transcriptomic profiles with a number of previously identified CSC and mammary stem cell (MaSC)-related gene sets[26,38,39]. The GSEA analysis showed that the gene sets upregulated in CSCs or MaSCs (IGS_UP, MET_CD44_UP, N_CD44_UP) were significantly enriched in the *PTX3*-overexpressing cells, while the gene sets downregulated in CSCs or MaSCs (MET_CD44_DOWN, MaSC_DOWN) were enriched in the control cells (Fig. 4i and Supplementary Fig. 2c), corroborating a role of PTX3 in CSC regulation.

Interestingly, GSEA analysis also revealed the upregulation of Hedgehog and Yes associated protein (YAP) signaling pathways after *PTX3* overexpression. The gene sets upregulated by Sonic Hedgehog (SHH_UP)[40] or its downstream transcription factor GLI1 (GLI1_UP)[41], and by YAP (YAP_UP)[42] were significantly enriched in *PTX3*-overexpressing cells as compared with control cells (Fig. 4j and Supplementary Fig. 2d). Upregulation of Hedgehog and YAP downstream genes after *PTX3* overexpression was also confirmed by qRT-PCR (Fig. 4k). Previously many studies have showed that activated Hedgehog[43] and YAP[44], which is the transcription factor effector of Hippo pathway, play essential roles in the regulation of cancer stemness. Thus, our data

support a functional role of PTX3 in BCSCs, likely by regulating the Hedgehog and Hippo-YAP pathways.

**PTX3 mediates the role of SH3RF3 in BCSC regulation.** Based on the above observations that *SH3RF3* enhances the expression of *PTX3* and that *PTX3* plays a role in BCSC regulation, we hypothesized that *PTX3* is the main downstream gene of SH3RF3 to mediate its role in BCSCs. To test this hypothesis, we first analyzed whether the transcriptomic changes caused by *SH3RF3* overexpression and *PTX3* overexpression were similar. The ssGSEA analysis showed that the genes with differential expression by *SH3RF3* and *PTX3* overexpression were coordinately expressed in breast cancer cell lines (Fig. 5a). In addition, the genes sets enriched in *PTX3*-overexpressing cells were also enriched after *SH3RF3* overexpression (Fig. 5b). Furthermore, the Hedgehog and Hippo-YAP downstream genes that were regulated by *PTX3* were also regulated by *SH3RF3* overexpression (Fig. 5c). These data indicated that SH3RF3 and PTX3 result in consistent transcriptomic effects of cancer cells, especially on the genes related to cancer stemness.

Next we tested whether PTX3 inhibition is able to rescue the effect of SH3RF3 on CSCs by siRNA silencing of *PTX3* in control and *SH3RF3*-overexpressing MDA-MB-231 cells (Fig. 5d and Supplementary Fig. 3a). Flow cytometry analyses revealed that *SH3RF3* expanded the ALDH+ CSC subpopulation in MDA-MB-231, while *PTX3* inhibition shrank the CSC population and effectively blocked the effect of *SH3RF3* overexpression (Fig. 5e, f and Supplementary Fig. 3b). Furthermore, *PTX3* silencing impaired the promoting effect of *SH3RF3* on tumorsphere formation in both MDA-MB-231 and HMLE cells (Fig. 5g and Supplementary Fig. 3c). In addition, shRNA knockdown of *PTX3* in *SH3RF3*-overexpressing MDA-MB-231 cells restrained the

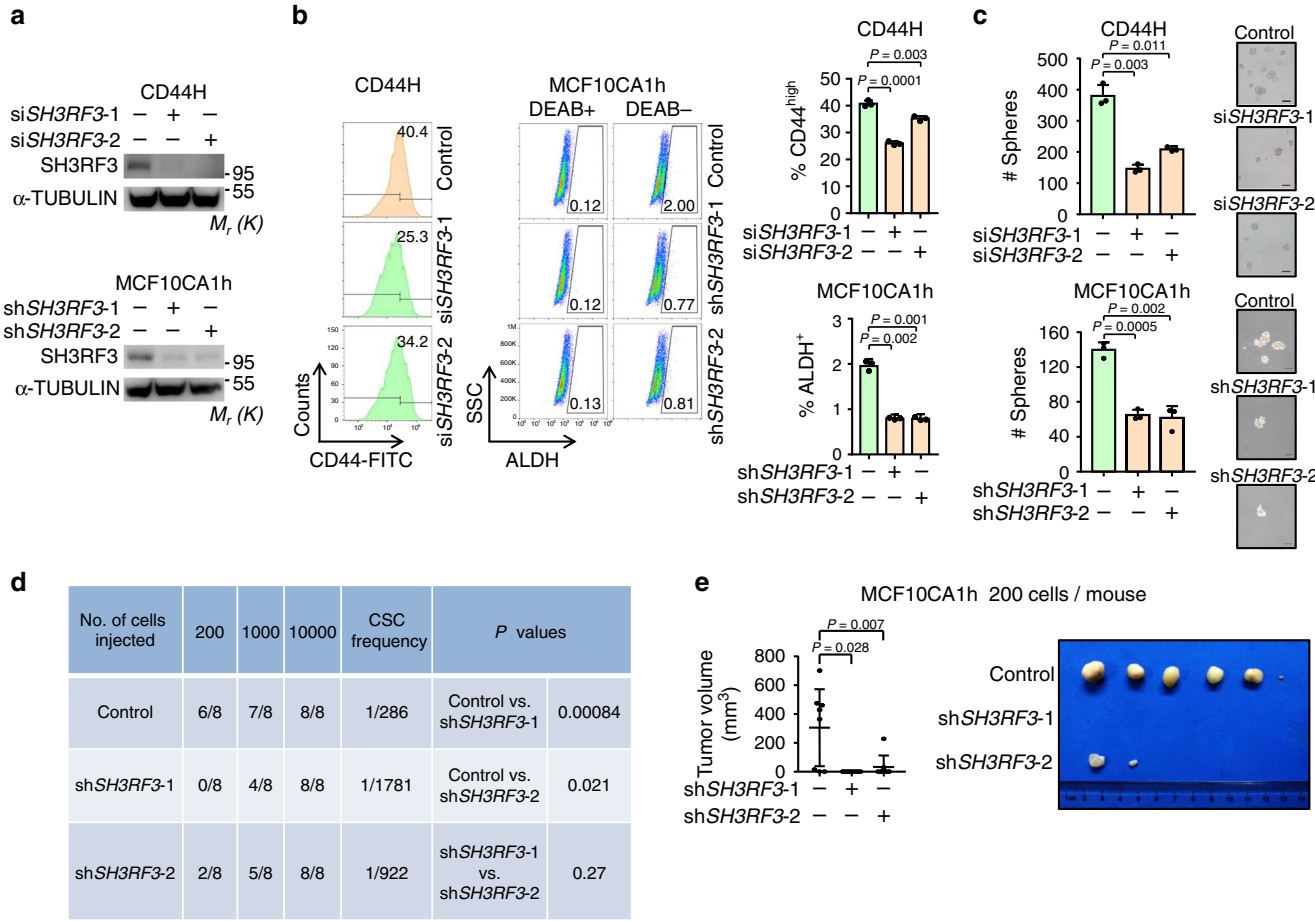

**Fig. 3 SH3RF3 knockdown impairs CSC traits of breast cancer cells. a** *SH3RF3* knockdown in HMLER-CD44H and MCF10CA1h cells. **b** Flow cytometry analyses of the CD44$^{high}$ subpopulation in HMLER-CD44H and the ALDH$^+$ subpopulation in MCF10CA1h ($n = 3$ cell culture experiments). **c** Quantitation and representative images of tumorsphere formation of breast cancer cells with *SH3RF3* knockdown ($n = 3$ sphere culture experiments). Scale bar, 100 μm. **d** In vivo tumor formation of the mice injected with serial dilutions of MCF10CA1h at day 42. **e** Tumor images and volumes ($n = 8$ mice) in mice injected with 200 control or *SH3RF3* knockdown MCF10CA1h cells. Data represent mean ± SD. Statistical significance was determined by two-tailed unpaired *t*-test (**b** and **c**), chi-squared test (**d**) or Mann–Whitney U-test (**e**). The experiments in (**a**), (**b**) (left), and (**c**) were repeated three times independently with similar results, and the data of one representative experiment are shown. Source data are provided as a Source data file.

enhancement of in vivo tumorigenesis by *SH3RF3* (Fig. 5h). These results collectively indicated that PTX3 acts at the downstream of SH3RF3 to promote CSC features of breast cancer cells.

**SH3RF3 enhances *PTX3* expression through JNK-JUN pathway.** To investigate how SH3RF3 enhances *PTX3* expression, we tested whether several CSC-related signaling pathways, including PI3K, JNK, WNT, and NF-κB, were involved in SH3RF3 regulation of *PTX3* expression. First, HMLE cells were treated with a PI3K inhibitor BKM120. It was found that BKM120 inhibited PTX3 expression and tumorsphere formation (Supplementary Fig. 4a and b), consistent to the previous report showing that *PTX3* was a target gene regulated by PI3K-AKT signaling[45]. However, *SH3RF3* overexpression was still able to upregulate PTX3 expression (Supplementary Fig. 4a) and tumorsphere formation (Supplementary Fig. 4b) in the presence of PI3K inhibition. Importantly, *SH3RF3* caused no difference in AKT phosphorylation (Supplementary Fig. 4c), further suggesting that SH3RF3 regulates PTX3 in a PI3K-independent pathway. In addition, we found that the JNK inhibitor (SP600125), but not the WNT or NF-κB inhibitors (XAV939 and BAY11-7082), could dramatically suppress the expression level of *PTX3* and eliminate the effect of SH3RF3 in HMLE cells (Fig. 6a). In addition, only

SP600125 was able to completely block the promotion of tumorsphere formation by SH3RF3, although the other inhibitors also partially reduced tumorspheres (Fig. 6b). The rescue effect of SP600125 on SH3RF3-regulated *PTX3* expression and tumorsphere formation was also observed in MDA-MB-231 cells (Fig. 6c, d).

Thus we further investigated the effect of SH3RF3 on the JNK signaling pathway, and found that *SH3RF3* overexpression led to increased phosphorylation of JNK1, JNK2, and the downstream transcription factor JUN, while *SH3RF3* knockdown inhibited JNK and JUN phosphorylation in multiple cell lines (Fig. 6e and Supplementary Fig. 4d). In addition, a list of known JUN-target genes, including MMP3, CD44, EGFR, ENPP2[46], and MGST1[47], were markedly upregulated in cancer cells with *SH3RF3* overexpression (Fig. 6f). These results suggested that *SH3RF3* regulates *PTX3* expression by activating the JNK pathway to promote BCSC properties.

Further, we studied the mechanism of *PTX3* regulation by JNK signaling. JUN is the transcription factor downstream of JNK and activates target gene expression by forming the AP-1 heterodimer complex. Chromatin-immunoprecipitation (ChIP)-qPCR assays revealed the binding of JUN on *PTX3* promoter (Fig. 6g and Supplementary Fig. 4e). In addition, JUN could directly activate the transcriptional activity of *PTX3* promoter, as shown by

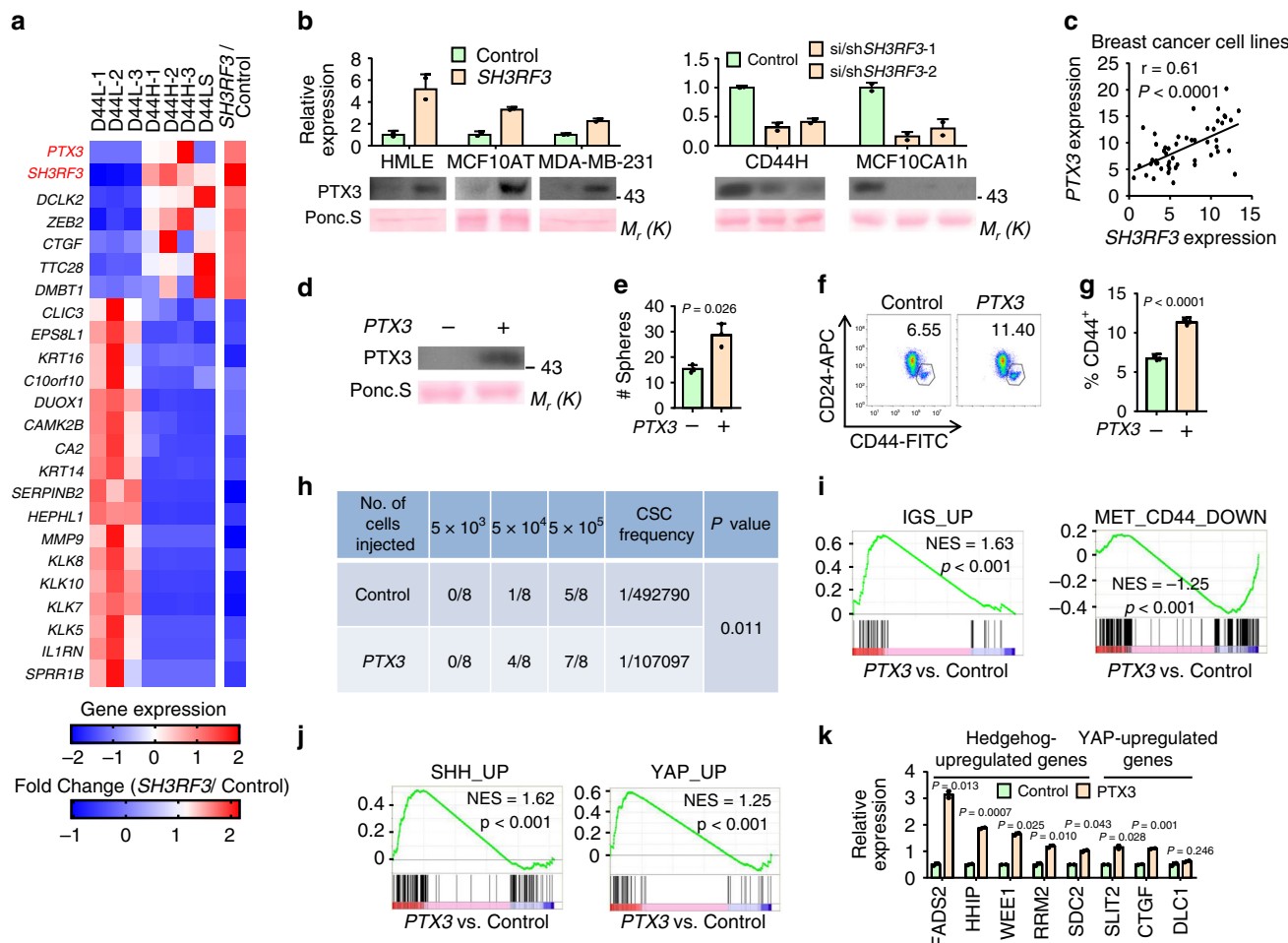

**Fig. 4 PTX3 is regulated by *SH3RF3* and enhance BCSC properties. a** Expression heatmap of the genes differentially expressed in HMLER sublines and regulated by *SH3RF3* overexpression. **b** *PTX3* mRNA expression and its extracellular protein levels in conditioned media of various cancer cell lines after *SH3RF3* overexpression and knockdown (*n* = 3 cell culturing). **c** Correlation of *SH3RF3* and *PTX3* mRNA levels in breast cancer cell lines (*n* = 51 cell lines). **d** Extracellular PTX3 levels in conditioned media of *PTX3*-overexpressing and control HMLE cells. **e** Tumorsphere formation of HMLE after *PTX3* overexpression (*n* = 3 culturing experiments). **f**, **g** Flow cytometry analysis (**f**) and quantitation (**g**) of CD44$^+$CD24$^-$ CSC subpopulation in *PTX3*-overexpressing and control HMLE cells. Scale bar, 100 μm; *n* = 3 cell culturing experiments. **h** Tumor incidence in the mice injected with various numbers of HMLER cells at day 90. **i** GSEA analyses of CSC and MaSC-related gene sets in *PTX3*-overexpressing versus control cells. **j** GSEA analyses of Hedgehog and Hippo-regulated gene sets in *PTX3*-overexpressing versus control cells. **k** mRNA levels of Hedgehog and Hippo-regulated genes after *PTX3* overexpression (*n* = 3 cell culturing experiments). Data represent mean ± SD. Statistical significance was determined by paired *t*-test (**c**) or two-tailed unpaired *t*-test (**e**, **g**, and **k**). The experiments in (**b**), (**d**), (**e**), (**f**), and (**k**) were repeated three times independently with similar results, and the data of one representative experiment are shown. Source data are provided as a Source data file.

luciferase reporter assays (Fig. 6h). Analyses with serial truncations of PTX3 promoter showed that JUN mainly targeted the −206 to +52 region in the promoter (Fig. 6h). In this region three AP-1 binding sites were identified (Supplementary Fig. 4e). Mutating one or two of these binding sites partially reduced the promoter activation by JUN, while mutating all three sites completely eliminated the response to JUN (Fig. 6h). Thus, all the three sites contributed to the regulation of *PTX3* expression by JUN. Collectively, these data suggested that JUN activated by SH3RF3 binds to the promoter of *PTX3* to enhance its expression.

SH3RF3 contains multiple SH3 domains which mediate protein–protein interactions[9]. Therefore, SH3RF3 might regulate JNK phosphorylation by interacting with JNK and its kinases. Indeed, reciprocal co-immunoprecipitation (co-IP) assays showed that SH3RF3 interacted with JNK1 and JNK2 (Fig. 7a and Supplementary Fig. 5a). The binding of SH3RF3 with MAP kinase kinase 4 (MKK4) and 7 (MKK7), the two main JNK kinases, was also observed (Fig. 6b and Supplementary Fig. 5b).

The phosphorylation of JNK is coordinated in a complex of JNK and MKKs, with the JNK-interacting protein (JIPs) acting as the essential scaffold for the complex organization[48]. Thus we studied whether the interaction between SH3RF3 and JNK in breast cancer cells is dependent on JIPs. We found that SH3RF3 could interact with all three JIP family members JIP1, JIP2, and JIP3 in 293T (Fig. 7c and Supplementary Fig. 5c); however, *JIP3* was the only JIP gene expressed in breast cancer cells MCF10AT (Supplementary Fig. 5d). In addition, sequential IP analysis proved the presence of SH3RF3, MKK7, and JIP3 in one complex (Fig. 7d). Interestingly, SH3RF3 enhanced the interaction of JIP3 to MKK7, but had no effect on JIP3-JNK2 interaction (Fig. 7e and Supplementary Fig. 5e). Instead, *JIP3* knockdown by two independent siRNAs (Supplementary Fig. 5f) obviously impaired the binding of SH3RF3 to JNK proteins (Fig. 7f) and SH3RF3-promoted JUN phosphorylation in MCF10AT cells (Fig. 7g). We further analyzed SH3RF3 protein region to interact with JNK signaling complex components and found that it was the fourth

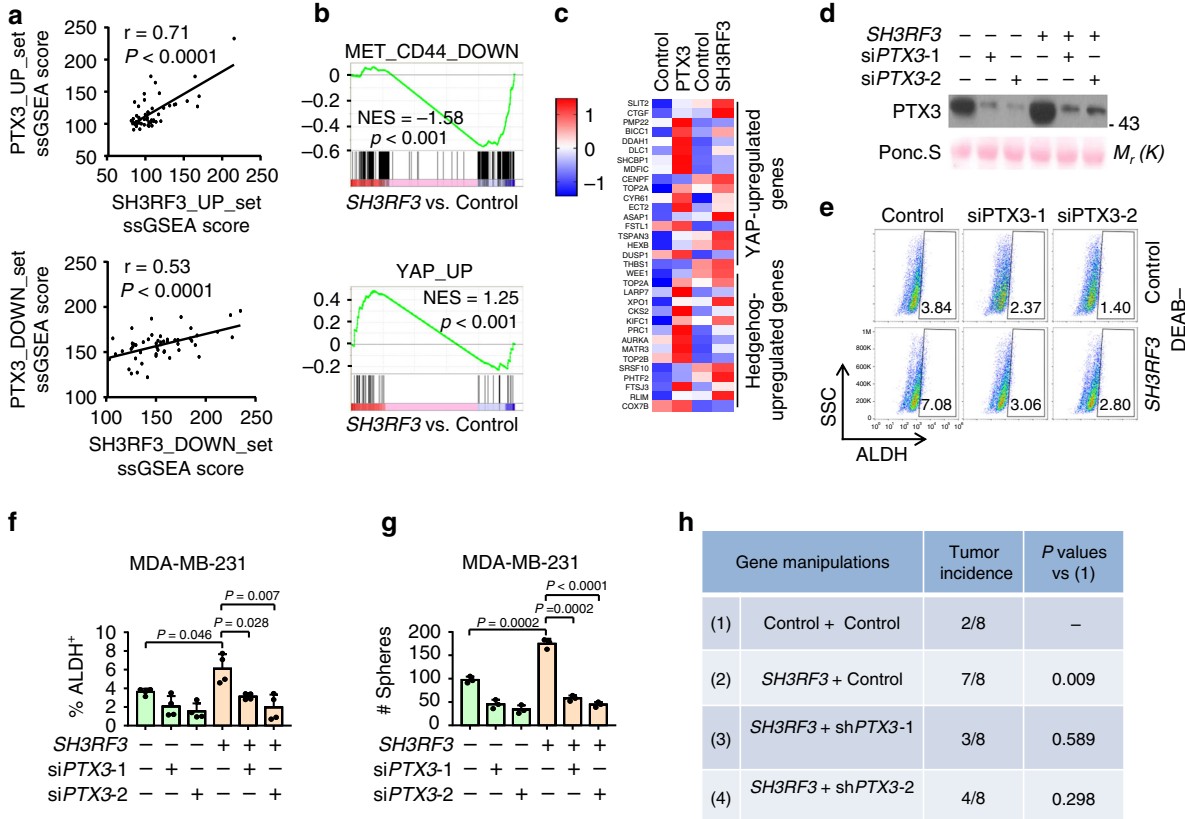

**Fig. 5 PTX3 mediates the function of SH3RF3 in BCSC regulation. a** Correlation of ssGSEA scores of the gene sets regulated after *PTX3* and *SH3RF3* overexpression in breast cancer cell lines ($n = 51$ cell lines). **b** GSEA analyses of CD44 or YAP-related gene sets in *PTX3*-overexpressing versus control cells. **c** Expression heatmap of Hedgehog and Hippo-regulated genes after *PTX3* and *SH3RF3* overexpression. **d** *PTX3* knockdown in MDA-MB-231 with *SH3RF3* overexpression. **e** Flow cytometry analyses of ALDH$^+$ subpopulation in MDA-MB-231 with *SH3RF3* overexpression and *PTX3* knockdown. **f** Quantitation ($n = 4$ FACS experiments) of ALDH + CSC subpopulations in MDA-MB-231 with SH3RF3 overexpression and PTX3 knockdown. **g** Tumorsphere quantitation in MDA-MB-231 with *SH3RF3* overexpression and *PTX3* knockdown ($n = 3$ culturing experiments). **h** Tumor initation in the mice injected with 40 cells of MDA-MB-231 cells at day 28. Data represent mean ± SD, $n = 3$. Statistical significance was determined by paired $t$-test (**a**) or two-tailed unpaired $t$-test (**f** and **g**). The experiments in (**d**), (**e**), and (**g**) were repeated three times independently with similar results, and the data of one representative experiment are shown. Source data are provided as a Source data file.

SH3 domain that interacts with MKK7, while the first and second SH3 domains interact with JIP3 and JNK1 (Fig. 7h and Supplementary Fig. 5g). These results suggested that SH3RF3 facilitates the assembly of MKK-JNK complex in a JIP-dependent manner and promotes JNK-JUN activation.

**SH3RF3 correlates with CSC properties in clinical tumors.** To assess the clinical significance of SH3RF3 in CSC regulation, we analyzed the functional role of SH3RF3 in patient-derived organoids, a 3D-culture CSC assay with high clinical relevance[49]. Patient-derived tumor cells of three breast cancers were infected with control or *SH3RF3*-expressing retroviruses and seeded in matrigel with organoid culture media. Notably, *SH3RF3* overexpression resulted in formation of more organoids in all three tumors, and larger sizes of the organoids (Fig. 8a, b). Similar results were observed in the organoid culture of four human gastric cancers (Fig. 8c), corroborating a role of SH3RF3 to promote CSC features of clinical tumor cells.

We then analyzed the RNA-sequencing data of the TCGA breast cancer clinical cohort. In these tumors, the expression of *SH3RF3* positively correlated to *PTX3*, as well as the CSC markers *CD44* and *ALDH1A1/2/3* (Fig. 8d). Significant correlation of *PTX3* to *CD44* and *ALDH1A1/2/3* was also observed (Fig. 8d). Moreover, ssGSEA analyses showed that the expression of

*SH3RF3* was positively correlated with the enrichment of CSC and MaSC-related gene sets in the TCGA cohort (Fig. 8e).

Next, we assessed the expression of SH3RF3 and CD44 at the protein level in a Qilu cohort of 40 breast invasive ductal carcinomas by immunofluorescence (IF) staining. The samples were divided into three groups according to the SH3RF3 IF intensities, and significantly more CD44$^+$ cells were found in tumors with higher SH3RF3 expression (Fig. 8f, g). Meanwhile, a positive correlation between SH3RF3 signal intensities and CD44 signal intensities was also noticed in these samples (Fig. 8h). Finally, we observed a link of *SH3RF3* expression to increased risk of distant metastasis (Supplementary Fig. 6a) and lower survival (Fig. 8i) of breast cancer patients in the Kaplan–Meier Plotter database[50]. Higher PTX3 protein expression was also associated with disease recurrence in the CPTAC breast cancer cohort (Supplementary Fig. 6b). In summary, our data confirmed that SH3RF3 plays a positive role in BCSC maintenance and is clinically associated with CSC enrichment and poor prognosis of breast cancer.

## Discussion
SH3RF3 belongs to a protein family characterized with the presence of multiple SH3 domains and a Ring finger domain. In addition to SH3RF3, there are two other members, SH3RF1

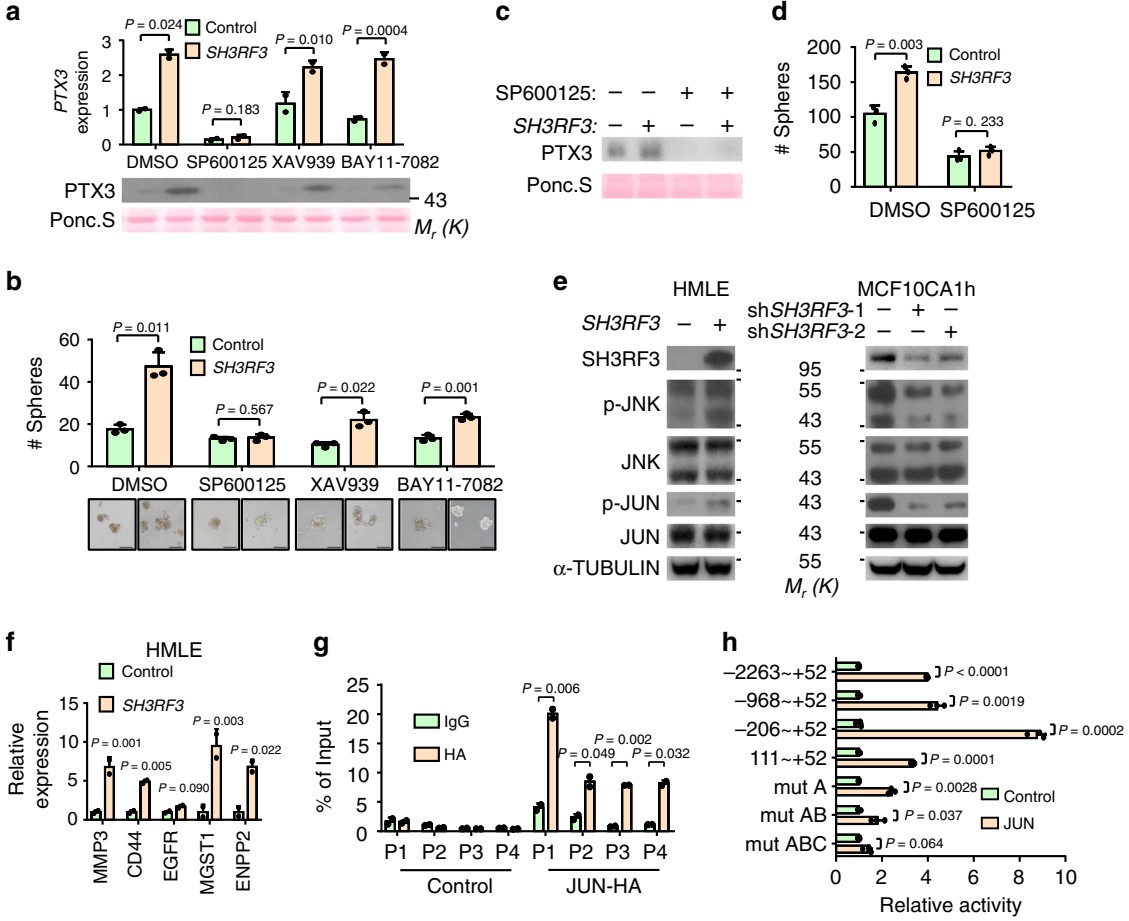

**Fig. 6 SH3RF3 enhances PTX3 expression through JNK-JUN pathway. a**, **b** *PTX3* expression (**a**) and tumorsphere formation (**b**, $n = 3$ culturing experiments) of *SH3RF3*-overexpressing and control HMLE cells treated with inhibitors of JNK (SP600125, 5 nM), WNT (XAV939, 5 nM), and NF-κB (BAY11-7082, 5 nM). **c**, **d** PTX3 expression (**c**) and tumorsphere formation (**d**, $n = 3$ culturing experiments) of *SH3RF3*-overexpressing and control MDA-MB-231 cells treated with the JNK inhibitor. **e** JNK and JUN phosphorylation after *SH3RF3* overexpression and knockdown. **f** Expression of JUN-target genes after *SH3RF3* overexpression in HMLE ($n = 3$ qRT-PCR experiments). **g** ChIP-qPCR analysis of JUN binding to *PTX3* promoter in HeLa ($n = 3$ ChIP experiments). **h** Luciferase reporter analysis of *PTX3* promoter in *JUN*-overexpressing and control HeLa cells ($n = 3$ reporter assays). Data represent mean ± SD. Statistical significance was determined by two-tailed unpaired $t$-test (**a**, **b**, **d**, **f**, **g**, and **h**). The experiments in (**a**–**f**) and (**g**) were repeated three times independently with similar results, and the data of one representative experiment are shown. Source data are provided as a Source data file.

(POSH) and SH3RF2 (POSHER), in the family. Among these proteins, SH3RF1 has been relatively well studied, with previous reports showing its roles to promote cell apoptosis and regulate various pathological processes including HIV infection and cerebral ischemia[51–53]. SH3RF2 has been reported to inhibit cell apoptosis by antagonizing SH3RF1[54]. However, the roles of these POSH members in cancer are unexplored, although dysregulation of *SH3RF1* and *SH3RF3* has been implicated in lung cancer and leukemia[11,55]. Here we report a previously unidentified role of SH3RF3 in stemness regulation of breast cancer. By transcriptomic screening of clinical tumors and cancer cell lines, we found that the expression of *SH3RF3* was consistently correlated with CSC properties of breast cancer cells. More importantly SH3RF3 not only promotes, but also is required to maintain the stem-like features of cancer cells. We also demonstrate that SH3RF3 contributes to the assembly of MKK-JIP-JNK complex, leading to activation of JNK and upregulation of *PTX3* expression for CSC regulation in cancer cells (Fig. 8j). The effect of SH3RF3 on the JNK pathway is similar to SH3RF1, which is also a JNK activator[53], but opposite to the JNK-suppressing SH3RF2[54]. This is in parallel to the structural homology of these proteins in that SH3RF2 only has three SH3 domains and our observation that the fourth SH3 domain mediates the interaction of SH3RF3 to the

JNK kinase MKK. Although the detailed mechanism of SH3RF3 to mediate the interaction of JNK and MKK is yet to be delineated, our study will expand our understanding of the POSH family proteins and CSC regulation in cancer.

In our analysis of HMLER cells, consecutive tumorsphere culturing of the non-CSC CD44L subpopulation resulted in a new subline with CSC traits including enhanced tumorsphere formation and chemoresistance. This spontaneous conversion of neoplastic non-stem cells to CSCs is concordant to previous reports[56] and highlights the plasticity of cancer cells, which is a key reason for the ultimate failure of many of current cancer therapeutics. Notably, previous studies showed that the spontaneous acquisition of CSC features of cancer cells was usually accompanied with the EMT process. In our study, although the initial CD44L and CD44H subpopulations of HMLER indeed displayed the epithelial and mesenchymal-like morphology, respectively, the CD44LS cells maintained an epithelial morphology (Fig. 1a), indicating EMT-independent mechanisms of inter-conversion between non-CSCs and CSCs.

BCSCs were also characterized by heterogeneity. A previous study demonstrated that an epithelial-like subpopulation and a mesenchymal-like subpopulation, labeled by $ALDH^+$ and $CD44^+CD24^-$, respectively, exist in BCSCs[24]. In our study, although

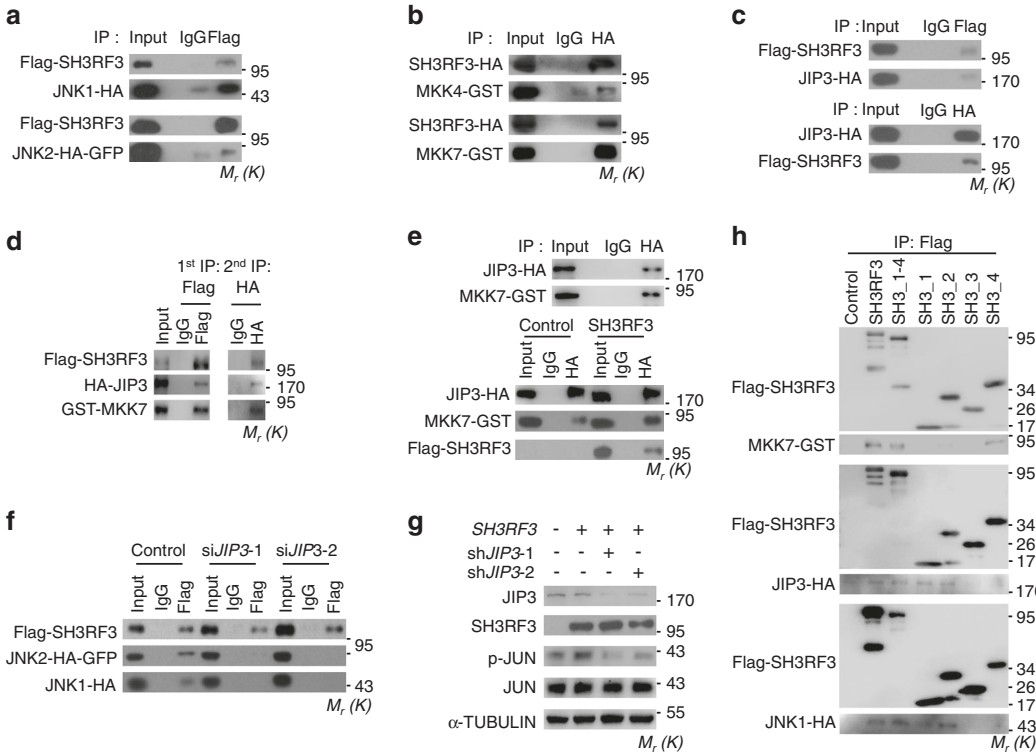

**Fig. 7 SH3RF3 activates JUN by interacting with MKK and JNK. a** Co-IP analyses of SH3RF3-JNK1/2 interaction in 293T cells. **b** Co-IP analyses of SH3RF3-MKK4/7 interaction in 293T cells. **c** Co-IP analyses of SH3RF3-JIP3 interaction in 293T cells. **d** Sequential Co-IP analyses of SH3RF3-JIP3-MKK7 interaction in 293T cells. **e** Co-IP analyses of JIP3-MKK7 interaction in MCF10AT (top), and with or without *SH3RF3* overexpression (bottom). **f** Co-IP analyses of SH3RF3-JNK1/2 interaction in MCF10AT with or without *JIP3* knockdown. **g** JUN phosphorylation after *SH3RF3* overexpression and *JIP3* knockdown in MCF10AT. **h** Co-IP analyses of the interactions of various SH3RF3 truncations to MKK7, JIP3, and JNK1. The experiments were repeated three times independently with similar results, and the data of one representative experiment are shown. Source data are provided as a Source data file.

both the CD44H and CD44LS sublines were characterized by enhanced stemness, CD44H was mainly CD44+CD24− and only contained a minor fraction of ALDH+ cells. In contrast, there was an obvious shift of the CD44LS population to higher ALDH expression (Fig. 1c). This observation corroborates the notion of CSC heterogeneity in breast cancer. More importantly, SH3RF3 can promote the expansion of both subtypes of BCSCs in different cell lines (Figs. 2b and 3b). Although different subtypes of BCSCs share little overlap and display distinct molecular profiles[24], SH3RF3 and JNK signaling might represent a common regulatory mechanism of heterogeneous BCSCs. Therefore, these results might have important implications for CSC-targeting designing in breast cancer therapeutics.

The JNK pathway is known to be involved in inflammation, apoptosis, cell differentiation, and proliferation. Recently, multiple studies have reported a critical role of JNK in CSC regulation[57]. Furthermore, JNK impairment led to inhibition of the stem properties of cancer cells in colorectal cancer and breast cancer[57–59]. However, the mechanism of JNK activation in CSCs is still largely unknown. JNK phosphorylation and activation by MKK is dependent on the formation of the JNK signaling complex that is composed of JNK, MKK and the scaffold JIPs. Here we showed that in breast cancer cells SH3RF3 also plays a role in the complex and contributes to JNK phosphorylation. This discovery will enrich our understanding on JNK signaling, and it warrants further investigation whether SH3RF3 is a genuine component of JNK signaling complex or its role in JNK and CSC regulation is context-specific in cancer cells.

Previous reports have firmly established the role of PTX3 to promote dedifferentiation[60], chemotherapy resistance[61], and stem properties of basal-like breast cancer cells[45]. However, there are

also studies reporting the silence of *PTX3* expression in tumors as compared with normal tissues[18,19] and its onco-suppressing effects in multiple cancer types[18,62,63]. These seemingly contradictory observations could be reconciled by the fact that *PTX3* was highly expressed only in CSC subpopulation of breast cancer and was nearly undetectable in non-CSCs (Fig. 4a and Supplementary Data File 1). Given that CSCs typically compose only a minor fraction of tumor cells, upregulation of PTX3 in CSCs may not be inconsistent to its weak expression in bulk tumors. These data might also indicate the possibility that PTX3 may play different roles in CSCs and non-CSCs for tumor regulation. In addition, our data showed that PTX3 expression results in activation of Hedgehog signaling and suppression of Hippo signaling, which might shed light on the unsolved question regarding the functional mechanism of PTX3 in cancer regulation. Nevertheless, the downstream molecular event of PTX3, especially the possible receptor of PTX3 to transmit the signal of the secretory protein into cells, is yet to be discovered before PTX3 targeting becomes practical for cancer treatment.

## Methods

**Plasmids and reagents.** *SH3RF3* cDNA was cloned into pRVPTO-puro-HA[64], pLVX-puro-Flag, and pLVX-blasticidin (Clonetech), respectively, for overexpression and Co-IP assays. PTX3, JNK1, JNK2 (fused with GFP), JIP1, JIP2, and JIP3 were cloned into pLVX-puro-HA, and MKK4/7 were cloned into pLVX-puro-GST for overexpression and Co-IP assays. For *SH3RF3* and *PTX3* knockdown, shRNAs were designed with the online software *sfold*[65], followed by annealing and cloning of the oligonucleotides into pSUPER-retro-puro (Oligo Engine) or pSU-PER-retro-blasticidin, with the following target sequences: 5′-ATTTCGAGA TGAAGGACAAAG-3′ (*SH3RF3* KD1); 5′-TCGAGGAAGGGCCACTATAAT-3′ (*SH3RF3* KD2); 5′-GGCCGAGAACTCGGATGATTA-3′ (*PTX3* KD1); 5′-GCC ATGGTGCTTTCAGTTTAA-3′ (*PTX3* KD2). Sequences for siRNA knockdown, including *SH3RF3* siRNA1 (5′-CCAAGAAACGCCACUCCUUTT-3′), siRNA2 (5′-

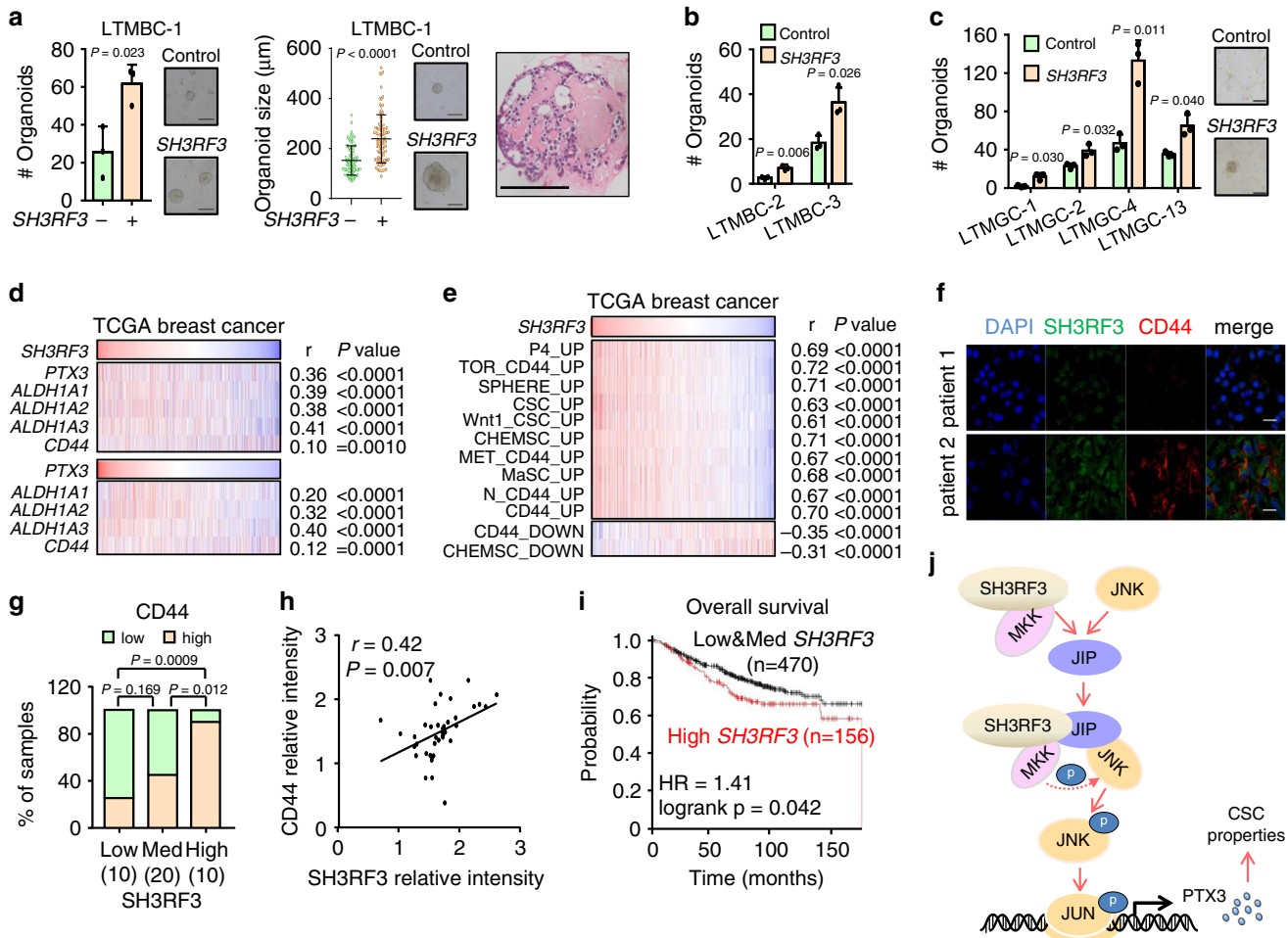

**Fig. 8 The clinical relevance of SH3RF3 in human breast tumors. a** Patient-derived organoid formation ($n = 3$ organoid culture replicates) in a human breast tumor, LTMBC-1, after *SH3RF3* overexpression. (Right panel) Lines represent median values and whiskers indicate the standard deviation. Haematoxylin & eosin staining of a representative organoid was shown on the right. Scale bar, 100 μm. **b** Patient-derived organoid formation ($n = 3$ organoid culture replicates) of two other human breast tumors (LTMBC-2 and 3) after *SH3RF3* overexpression. **c** Patient-derived organoid formation ($n = 3$ organoid culture replicates) of four human gastric cancers (LTMGC-1, 2, 4, and 13) with *SH3RF3* overexpression. Scale bar, 100 μm. **d** Correlation of *SH3RF3* and *PTX3* expression to different CSC marker genes in TCGA breast cancer cohort ($n = 1097$ patients). **e** Correlation of *SH3RF3* expression and the ssGSEA scores of CSC and MaSC-related gene sets in TCGA breast cancer cohort ($n = 1097$ patients). **f** Representative IF images of SH3RF3 and CD44 in breast cancer tissues of the Qilu cohort. Scale bar, 100 μm. **g** CD44 expression levels in breast cancer tissues of the Qilu cohort with different SH3RF3 expression ($n = 40$ patients). Numbers in parenthesis indicate sample sizes. **h** Correlation of SH3RF3 and CD44 IF intensities in breast cancer tissues of the Qilu cohort ($n = 40$ patients). **i** Survival analysis by *SH3RF3* expression in Kaplan–Meier Plotter breast cancer cohort ($n = 626$ patients). **j** Schematic model of the role of SH3RF3 in BCSC regulation. Data represent mean ± SD. Statistical significance was determined by two-tailed unpaired *t*-test (**a**, **b** and **c**), chi-squared test (**g**), paired *t*-test (**h**) or two-sided log-rank test (**i**). Source data are provided as a Source Data file.

CCCAACGUCAGUGCCGCAATT-3′), *PTX3* siRNA1 (5′-GCACAAAGAGGA AUCCAUAUU-3′), siRNA2 (5′-GGGAUAGUGUUCUUAGCAATT-3′), *JIP3* siRNA1 (5′-GCGGCACACAGAGAUGAUAUU-3′), siRNA2 (5′-CCAGCACCA CAGGCACCAATT-3′), and the scrambled control (5′-UUCUCCGAACGUGU CACGUUU-3′) were purchased from GenePharma.

For *PTX3* promoter activity analysis, the −2263 to +52 DNA fragment of the *PTX3* gene locus, as well as the truncated and mutated sequences, was cloned into pGL3-basic (Promega) with MluI and BglII sites. All primers used in this study are provided in Supplementary Table 3. All the constructs were verified by DNA sequencing. All cell lines used in this study were tested as mycoplasma free.

For Western blotting, flow cytometry, Co-IP and IF analyses, the following antibodies were used: rabbit anti-human SH3RF3 (ThermoFisher, PA560841), rabbit anti-PTX3 (Proteintech, 13797-1-AP), rabbit anti-human JIP3 (Proteintech, 25212-1-AP), rabbit anti-human JNK2 (Proteintech, 51153-1-AP), rabbit anti-human phospho-JUN (Ser73) (Cell Signaling Technology, 9164S), rabbit anti-human JUN (Cell Signaling Technology, 9165P), rabbit anti-human phospho-SAPK/JNK (Thr183/Tyr185) (Cell Signaling Technology, 4668t), APC mouse anti-human CD24 (Biolegend, 311118), FITC mouse anti-human CD24 (BD Pharmingen, 555478), PE mouse anti-human CD44 (BD Pharmingen, 555479), Alexa Fluor 488 donkey anti-rabbit IgG (Invitrogen, A-21206), Alexa Fluor 647 goat anti-rat IgG (Biolegend, 405416), rabbit anti-human GAPDH (Sigma, G9545),

mouse anti-human α-TUBULIN (Sigma, T6199), mouse anti-human β-ACTIN (Sigma, A2228), rat anti-human CD44 (Santa Cruz, sc-18849), mouse anti-HA (Sigma, H9658), mouse anti-Flag (Sigma, F1804), rabbit anti-HA (Cell Signaling Technology, 3724S). All antibodies were diluted according to the dilution rate recommended in the manufacturers' instructions before use. The inhibitors of various signaling pathways used in this study were BKM120 (MCE, HY-70063), SP600125 (Merck, 420119), BAY11-7082 (Selleck, s2913), and XAV939 (Sigma, X3004).

**RNA sequencing.** Total RNA was extracted from the cells using the Trizol reagent (Invitrogen), and evaluated with a 2100 Bioanalyzer (Aglient Technologies). RNA samples with an RNA integrity number (RIN) > 8 were used for subsequent sequencing. Library construction and sequencing were performed at WuXi Next-CODE, Shanghai. mRNA was enriched from total RNA samples by rRNA removal (HMLER CD44L, CD44H, CD44LS subline cells) or PolyA purification (*SH3RF3*-overexpressing MCF10AT and *PTX3*-overexpressing HMLER cells), and then fragmented, followed by reverse transcription with random primers. Then the synthesized cDNA was subjected to end-repair, phosphorylation and 'A' base addition according to Illumina's library construction protocol. After Illumina sequencing adapter addition, PCR amplification and purification with AMPure XP Beads (Beckmen), sequencing was performed with a HiSeq X10 sequencing

platform (Illumina) following Illumina-provided protocols, resulted in 2 × 150 bp paired-end reads. Skewer v0.2.2[66] was used to remove the adapter sequences. Quality control (QC) was performed to remove bases with base quality <Q30 and reads with length < = 75 bp from subsequent analysis. Data yields of the analyzed samples in this study were 53.90–75.51 M raw reads and 53.81–75.28 M QC-passed reads. Then the filtered reads were mapped to the human genome (hg19) and transcriptome (gencode v19) using the software STAR v2.5.1b[67] with default parameters ("-outFilterMultimapNmax 10, -outFilterMismatchNmax 10, -outFilterMismatchNoverLmax 0.3, -sjdbOverhang 100, -outFilterType Normal -alignSJoverhangMin 5, -alignSJDBoverhangMin 3, -alignIntronMin 21, -alignIntronMax 0, -alignMatesGapMax 0"), yielding 93.7–97.3% alignment of the reads. PCR duplication rates of the RNA-sequencing data in this study, estimated by dupRadar[68], were 0.303 ± 0.118 for all the samples. The read duplicates were retained in the following expression analysis in that it was hard to distinguish PCR duplicates from nature duplicates in this sequencing protocol. Expression estimation was performed using RSEM 1.2.29[69] with default parameters except the setting "--estimate-rspd". Across-sample normalization and differential expression analysis were performed using the R package EdgeR v3.8.5[70] with the trimmed mean of M-values algorithm. Possible multi-mapping of *SH3RF3* reads to the whole genome was also analyzed, and it was found that there were 0.6% *SH3RF3* reads multi-mapped to other genes, but none to the other two SH3RF family members, *SH3RF1* and *SH3RF2*. Genes with CPM (count per million) ≥5 (HMLER CD44L, CD44H, CD44LS subline cells) or CPM ≥ 1 (*SH3RF3*-overexpressing MCF10AT and *PTX3*-overexpressing HMLER cells) in at least one sample were taken into further analysis. Pairwise comparison between sample groups was performed with exact test for differential expression analysis. Up- and downregulated genes with fold changes >2 and Benjamini–Hochberg adjusted *p* values <0.05 were identified.

**Bioinformatic analyses of public transcriptomic datasets.** The GDC TCGA Breast Cancer RNAseq (HTSeq-FPKM-UQ) expression dataset was obtained from UCSC Xena[71]. The CSC enrichment of these TCGA tumors were analyzed by ssGESA[25] with the previously identified gene signature differentially expressed in CD44+CD24− tumor cells as compared with CD44−CD24+ counterparts (CD44_UP_DOWN, [26]). Correlation of the ssGSEA scores and expression of each gene was assessed by Pearson correlation analysis. The genes were ranked by Pearson's coefficient (*r*) and the top 100 positively and negatively correlated genes were used for further analysis. The RNA microarray data of patient-derived tumorspheres and parental tumors were downloaded from the GEO database (GSE7515), and the genes with fold changes >2 and Student's *t*-test *P* < 0.05 were selected for further analysis. The expression dataset of breast cancer cell lines was obtained from Cancer Cell Line Encyclopedia[37] and analyzed with ssGSEA. The National Cancer Institute CPTAC (Clinical Proteomic Tumor Analysis Consortium) breast cancer proteomics data were downloaded from the cBioPortal database[72,73] and used to analyze the correlation of PTX3 expression with cancer prognosis. The full list and description of the CSC or MaSC-related gene sets used in GSEA and ssGSEA were provided in Supplementary Data File 2.

**Flow cytometry analyses.** Cancer cells harvested from culture during the logarithmic growth period were resuspended in PBS. 500,000 cells were incubated with the antibodies in recommended concentrations by the manufacturer's instructions at 4 °C for 0.5 h, followed by three times of PBS washing. Then cells were filtered with a 70-μm strainer and sorted by a MoFlo Astrios EQ Flow Cytometer (Beckman) or analyzed by a Gallios Analyzer (Beckman). ALDH activity analysis was performed by the ALDEFLUOR Kit (StemCell Technologies) following the manufacturer's protocol. The specific ALDH inhibitor Diethylaminobenzaldehyde (DEAB) was used as negative control in ALDH analyses. Flow cytometry data processing was performed with FlowJov10 (Tree Star, USA). Flow cytometry gating strategy is shown in Supplementary Fig. 7.

**Tumorsphere formation.** Cells were cultured in serum-free medium with cell line-specific annexing agents (Supplementary Table 2). 5000–10,000 cells were seeded in 6-well ultra-low attachment plates (#3471, Corning, USA) and cultured for 1 or 2 weeks. The spheres with diameters larger than 50 μm were counted. The detailed information of tumorsphere culture conditions, including culture media, seeded cell numbers, and culture time, of different cell lines was provided in Supplementary Table 2. The culturing condition of CD44L consecutive tumorsphere passages for CD44LS derivation was the same for the initial tumorsphere culture.

**Chemoresistance assays.** Five thousand tumor cells were seeded in each well of a 96-well plate. Chemotherapy drugs in various concentrations (paclitaxel 0–200 ng/mL, doxorubicin 0–500 nM) or the solvent as the negative control were added into the wells. Cells were cultured for 48 h, followed by MTT assays of cell viability. 3-(4,5-dimethyl-2-thiazolyl)-2,5-diphenyl tetrazolium bromide (MTT) was prepared in PBS at a concentration of 5 mg/mL and pH 7.4. Twenty microliters of MTT solution was added in each well. After 4 h, the crystal was dissolved in DMSO and the absorbance was measured at 490 nm.

**Promoter activity assay.** The *PTX3* promoter activity regulated by JUN was measured by luciferase reporter assays. Two hundred microgram control or JUN

plasmids, 200 μg PTX3 promoter Firefly luciferase reporter plasmid and 100 μg Renilla luciferase plasmid were transfected into HeLa cells in each well of 24-well plates. Cells were cultured for another 48 h and lysates were collected for luciferase activity analysis in firefly (25 mM glycylglycine, 15 mM potassium phosphate, 15 mM MgSO$_4$, 2 mM ATP, 10 mM DTT, and 1 mM D-luciferin, pH 7.8) and Renilla (0.5 M NaCl, 1 mM EDTA, 0.1 M potassium phosphate, 0.04% BSA and 2 μM coelenterazine, pH 7.4) luciferase assay buffer.

**Chromatin immunoprecipitation (ChIP).** ChIP assays for *PTX3* promoter binding by JUN were conducted in HeLa cells. Briefly, HeLa cells were transfected with JUN-overexpressing or control plasmids. Two days later, cells were crosslinked with 1% formaldehyde and quenched by 125 mM glycine. Cell nuclear lysate was sonicated and incubated with control IgG or anti-HA antibody for immunoprecipitation. The complex was captured and precipitated by agarose beads. Captured genomic DNA was reverse-crosslinked and purified by ethanol precipitation with Dr. GenTLE Precipitation Carrier (Takara). Purified genomic DNA was used for qPCR analysis.

**Co-immunoprecipitation (Co-IP).** Cells were transfected with overexpression plasmids 48 h before Co-IP. The cell lysate was incubated with control IgG or immunoprecipitating primary antibodies overnight at 4 °C. Then, another 2-h incubation was conducted with 25 μl protein A/G agarose beads (GE). Beads were washed by lysis buffer for five times and heated at 95 °C for 10 min. Western blotting was performed to detect the interaction between the concerned proteins.

**Immunofluorescence (IF) analysis.** Fixed tumor tissues were embedded in paraffin and sectioned into 10-μm slices. The sections were dewaxed with xylene and hydrated by a series of ethanol solutions of different concentrations, followed by antigen retrieval. The sections were incubated with SH3RF3 or CD44 primary antibodies overnight at 4 °C, followed by washing with PBS for three times and an incubation with fluorescent-labeled secondary antibodies and DAPI. After washing and sealing, the sections were analyzed by confocal microscopy (ZEISS) and ZEN blue edition software (ZEISS).

**Animal studies.** Female NSG mice aged at 4–6 weeks were purchased from Shanghai Model Organisms Center, Inc, maintained in the standard SPF animal house and used in all studies. For tumor transplantation, cancer cells harvested from cell culture were resuspended in PBS at a concentration of 1 × 10$^7$ cells/mL and diluted to specific concentrations. An incision was made in the abdomen, and the skin was recessed to locate the #4 mammary fat pad, into which cells were injected under a dissection microscope. Tumor incidence of the mice was determined by palpation and confirmed by surgically opening the mammary glands for tumor examination at the end point of experiments. The tumor was removed from the mice before the volumes were measured. All animal studies were conducted in accordance with Guidelines for the Care and Use of Laboratory Animals that is approved by the Institutional Biomedical Research Ethics Committee of Shanghai Institutes for Biological Sciences.

**Clinical samples and patient-derived organoid (PDO) culture.** Fresh and paraffin-embedded breast cancer tissues were obtained from Qilu Hospital of Shandong University (Ji'nan, China) with informed patient consent and approval from the Institutional Review Board. Fresh tumor tissues were harvested and transplanted into cleared mammary fat pads of NOD-SCID mice pretreated with CD122 neutralizing antibodies to establish patient-derived xenograft of breast cancers. Tumor cells were dissociated from the established xenografts within two passages and were infected with control or *SH3RF3*-overexpressing retroviruses. The infected cells were seeded in matrigel in 24-well plates and covered by organoid medium supplemented with 1:50 B27, 0.5 μg/mL Hydrocortisone, 20 ng/mL bFGF, 20 ng/mL EGF, 5 μg/mL insulin, and 2% FBS. After one week of culture, the numbers of organoids were counted.

**Statistical analyses.** Data analysis was performed using GraphPad Prism 7.0 (GraphPad Software, La Jolla, USA). Unless stated otherwise, two-tailed Student's *t*-test without assumption of equal variance was conducted to compare the data of different groups of in vitro and animal assays. Breast cancer patient survival was analyzed by log-rank test. Pearson correlation coefficient was used to assess the correlation between different data groups. The *p* values <0.05 were regarded as statistically significant. All imaging experiments were repeated at least three times, with similar results.

**Reporting summary.** Further information on research design is available in the Nature Research Reporting Summary linked to this article.

## Data availability

The RNA sequencing data of *SH3RF3* and *PTX3*-overexpressing cells were deposited in the NODE database (https://www.biosino.org/node) with project IDs OEP000303, OEP000304, and OEP000305 and GEO database (https://www.ncbi.nlm.nih.gov/gds/)

with project ID GSE130577. The source data underlying Figs. 1b, c, e, f; 2a–c, e; 3a–c; e; 4a–e, g, k; 5a, c, d, f, g; 6; 7; 8a–c, g, h; Supplementary Figs. 1b–e; 2a, b; 3a, c; 4b–d and 5 are provided as a Source Data file. All the other data supporting the findings of this study are available within the article and its Supplementary information files and also from the corresponding author upon reasonable request. A reporting summary for this article is available as a Supplementary Information file.

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

## Acknowledgements
We thank Yan Ji, Xiang Miao, Kai Wang, Yujia Zhai, Chengji Wang, Yiting Yuan, Fengtao Qian, Shuyang Yan, Tao Huang, Jun Li, and Zhonghui Weng at the Institute of Nutrition and Health core facilities for technical support. The study was funded by National Natural Science Foundation of China (81430070, 81661148048, 81725017, 81802647), Chinese Academy of Sciences (QYZDB-SSW-SMC013), Ministry of Science and Technology of China (2017YFA0103502), the Program of Shanghai Academic Research Leader (19XD1404500), and Natural Science Foundation of Heilongjiang Province of China (QC2017111).

## Author contributions
G.H. was responsible for experimental design, data analysis, and the supervision of this project. P.Z. and Y.L. performed most of experiments and mainly contributed to writing the paper. C.L., X.L., and M.C. performed the PDO assays. X.C. and G.W. contributed to quality control of RNA sequencing. Y.W., P.T., and X.Z. contributed to the immuno-fluorescence analysis of clinical samples. T.L. contributed to the collection and analysis of clinical samples.

## Competing interests
The authors declare no competing interests.
