## [Peer Review File · Nature Communications]

Reviewers' comments:

Reviewer #1 (Remarks to the Author); expert in expert in JNK signalling:

The paper by Zhang et al. describes that the scaffold protein POSH2 is upregulated in cancer stem-like cells (CSCs) promoting breast cancer stem-like properties. It seems that these effects are mediated by JNK activation and PTX3 upregulation.

As it was already published that POSH mediates JNK activation (Xu et al., 2003) while the third member of the same family, POSHER, induces its degradation (Wilhelm et al., 2013) the authors would need to study the molecular mechanism of these differences. To really result in a significant increase of our knowledge of this pathway, it would be necessary to clarify which SH3 domain is involved and if the binding is to JNK, MKK or JIP.

There are several points that need to improve in order to be published:

- The authors make clear that previous reports firmly established the role of PTX3 in cancer. For this reason, the authors need to check whether it could be a target for cancer treatment. They need to carry on experiments to see if the lack/reduction of PTX3 reduce the tumor size, similarly to their data in Figure 3 targeting POSH2.
- Evaluate the interaction between the component of the signaling network (JIP, JNK, POSH2, MKK)
- The use of Ponceau S staining of the nitrocellulose membranes is not acceptable as loading control for the western blots. Antibodies such as tubulin, GAPDH or similar must be employed.
- It is important to demonstrate reduced activity of JNK when the interaction between POSH2 and JNK is impaired due to JIP3 knockdown (Fig. 6H). This can be done by showing Jun phosphorylation or the expression of Jun-target genes.
- Tumor incidence in animal studies should be determined by observing lack of tumors after opening the mice. If this is only determined by external palpation, some very small tumors would not be taken into consideration, and incidence would be underestimated.
- For the flow cytometry analysis of ALDH+ population, the negative control of DEAB+ (ALDH inhibitor) should be included, if not, how do the authors really know that the positive population is the real one?
- Authors sort HMLER cells in 2 populations in function of their CD44 expression levels. However, the plots in figure 1D show that the level of CD44 in both populations are almost the same. Does CD44LS population increase the levels of CD44 over time?
- Authors directly relate POSH2 levels with CD44high in HMLE and MCF10AT cell lines and ALDH+ expression in MDA-MB-231. Did the authors measure ALDH+ expression in HMLE and MCF10AT and CD44 in MDA-MB-231?
- Did the authors perform limiting dilution assays in POSH2 overexpressing-HMLE cell line? Why did the authors specifically use MDA-MB-231 cell line for this experiment? These experiments need to be done also in HMLE cell.
- In the siRNA-POSH2 experiments, did the authors check the siRNA-POSH2 in the MDA-MB-231 cell line?
- Previously it has been shown that PI3K signalling target penta-3 to promote stemness in basal-like breast cancer (Thomas et al. 2017 Sci Signal. 21;10 (467)). Does POSH2 exert its functions, at least in part, through the PI3K pathway? Authors should investigate this possibility.
- The information about the details in tumorsphere experiment is very deficient. The author should include the cell concentration at the beginning and in the passages.

Minor points

- The authors indicate that one band of p-JNK Western blot corresponds to p-JNK1, while the other one corresponds to p-JNK2 (Fig. 6C). However, each band is p-JNK1/2 (two different splice variants of JNK1 and JNK2).
- In Figures 2 & 3, the tubulin western blot control is clearly overexposed. A shorter exposition (or less protein loaded) must be presented.

- The authors should show total JNK and Jun protein levels in Fig. 6C.
- In the MMTs assays (Figure1C) authors used paclitaxel amounts between 0-200ng per well, but which volume was used per well? Inhibitors might be indicated by concentration in order to allow reproductivity of the experiments
- The number of replicates is only included in some experiments. Without sample size the presented results don't mean anything. I suggest authors to include this information in their figures.

Some of the figures are too small and there are some spelling errors.

- "Tumoresphere" should be "tumorsphere" in line of page 5.
- "Overexpression" should be "overexpression" in line 7 of Figure legend 5.
- Line 12 of Methods section lacks the word "in" before "this study".

Reviewer #2 (Remarks to the Author); expert in cancer bioinformatics:

The study Zhang et al present uses a wide array of techniques to investigate the biology of CSCs in breast cancer specifically and cancer more broadly. While most of what is presented in the present manuscript is correct, and conclusions are solid, the manuscript has a severe deficit in some techniques in explaining the methods used. The authors need to be aware that any other scientist that would want to replicate their results need to have the VERY DETAILED and FULL methodology used to produce the presented results, which is not the case. This is reflected in the specific comments below, but not restricted to these:

Throughout all the text:

Minor: Official symbol of the gene is SH3RF3, not POSH2. This needs to be reversed throughout the text (including the title). All other mentioned genes in the manuscript should be also revised to ensure that official gene symbols are used, and not any aliases or old names.

Minor: The authors make several statements like "These results demonstrated", where "These results suggested" would be more appropriate, since other explanations beyond the scope of this study could explain the changes.

Introduction

Minor: Page 3 line 12: change testified to tested

Minor: Rephrase the PTX3 sentences, just jumps into it, but needs to start saying first why are they talking about it.

Results

Major: 3 replicas of CD44L (control) and 2 replicas of CD44H and CD44LS. At least 3 replicas of each treatment need to be done.

Minor: In the mice injected with 40 control cells, were the injection of tumorigenic cells also 40? Furthermore, the increase is "only" 20%, have they controlled in their analysis for the expression of the other already known over or under-expressed cancer genes? Is there a specific gene expression environment in which the overexpression of SH3RF3 is more tumorigenic?

Minor: The 24 genes resulting from the overlap of HMLER and MCF10AT does not imply that they are controlled by SH3RF3, only that they are co-up- or co-down-regulated together with SH3RF3. Not causality, just co-occurrence. As they verify by knockdown, PTX3 seems to be the case, but not necessarily the other 23 genes, rephrase.

Methods

Major: For their own RNAseq data (SH3RF3 and PTX3 over-expressing cell lines), they do not mention absolutely anything about who the data was analysed. They need to state which algorithm was used for reads alignment, how they controlled and/or how they addressed for multimapping of reads, what kind of normalisation they used prior to differential expression analyses, which software/packages they used for all this processed (with references) and which default parameters of these software/packages were changed (if any). Also, which kind of RNAseq was performed? Whole transcriptome or something else? Was the RNAseq paired end or single end? What length of read? How many millions of read per sample they obtained (both raw and after QC)? Which were

the QC criteria to remove reads after alignment to reference? Mapping quality? Multimapping? PCR duplicates? Also, were the reads aligned to the hg19 or hg38 of the reference?

Major: Once all this is cleared, they also need to assess how this RNAseq analysis correlates technically with other external analyses considered for the study (e.g. TCGA). Is the RNAseq analysis performed by the authors comparable and compatible with that of the TCGA and others?

Major: SH3RF3 has genes with similar sequences and implicated in the same processes SH3RF1 and SH3RF2 (as the authors clearly discuss). What is the rate of multimap between the reads that mapped to each of the three genes? How was read multimapping addressed between these 3 genes?

Reviewer #3 (Remarks to the Author); expert in CSC and breast cancer:

In the submitted manuscript, authors show that POSH2, a scaffold protein increases CSC properties of breast cancer cells through activation of JNK-JUN pathway and Pentaxrin 3. Authors through in vitro tumorsphere formation assay and in vivo experiments coupled with clinical cohort data show the relevance of POSH3 in CSC properties of breast cancer. The manuscript is very well written and communicates the idea convincingly. The role of POSH2 in breast cancer CSC is novel. Given the need to target CSCs to overcome chemo and radio resistant in patients, this study possess clinical relevance.

However, I would like to suggest few corrections before publication of the article:

Major corrections:

1. Throughout the manuscript, authors have used orthotropic mice models to show the role of POSH2 in CSC maintenance. It would interesting to show if POSH2 expression correlates with CSC numbers and function in spontaneous breast cancer models.
2. Authors show that inhibition of PTX3 in control and POSH2 over expressing MDA-MB-231 and HMLE cells inhibit the POSH2 promoted tumorsphere formation. However, no in vivo experiment has been provided to show that inhibiting PTX3 hampers CSC properties of breast cancer cells. It would be good if authors could compare the tumor initiation capability of PTX3 knockdown cells to that of POSH2 knockdown cells.
3. In Figure 7, authors have used 3 patient samples, to study the correlation between POSH2 expression and CD44 expression. The sample number is very low to state that there is a positive correlation between the two. Also, in experiments involving patient derived organoids, n=3 which is very less to convincingly demonstrate the positive relationship between POSH2 expression and organoid number.

Minor corrections:

1. In Figure 2A, authors show over expression of POSH2 in HMLE, MCF10AT and MDA-MB-231. Over expression in MCF10AT is minimal while in the other two, tubulin looks saturated. Authors can repeat these blots for better quality. Also, please specify if it is beta tubulin.
2. Figure 2B, upon over expression of POSH2 in HMLE and MCF10AT, authors show increase in CD44+CD24- population, while in MDA-MB-231, ALDH+ population is increased. What happens to CD44+CD24- population in MDA-MB-231?
3. In Figure 4B, authors show relative expression of PTX3 between control and POSH2 over expression. Control bars are not showing unit "1", so what is the baseline for "relative expression"?
4. Blot for PTX3 in MCF10AT in Figure 4B is not convincing.
5. In Figure 7I, authors show a positive correlation between POSH2 expression and decreased overall survival. Is that relationship true with high PTX3 expression?

Point-by-point responses to reviewers' comments

Reviewer #1 (Remarks to the Author); expert in expert in JNK signalling:

The paper by Zhang et al. describes that the scaffold protein POSH2 is upregulated in cancer stem-like cells (CSCs) promoting breast cancer stem-like properties. It seems that these effects are mediated by JNK activation and PTX3 upregulation.

As it was already published that POSH mediates JNK activation (Xu et al., 2003) while the third member of the same family, POSHER, induces its degradation (Wilhelm et al., 2013) the authors would need to study the molecular mechanism of these differences. To really result in a significant increase of our knowledge of this pathway, it would be necessary to clarify which SH3 domain is involved and if the binding is to JNK, MKK or JIP.

We appreciate this suggestion and have investigated the domains of POSH2 (the other name of POSH2, SH3RF3, was used in the revised manuscript, following the suggestion of reviewer #2) interacting with MKK7, JIP3 and JNK1 (Fig. 7H and S5G). The data showed that the fourth SH3 domain of POSH2/SH3RF3 interacted with MKK7, while it was the first and the second SH3 domains that interacted with both JIP3 and JNK1. Further, we compared the protein structures of the 3 POSH family members and found that POSHER lack the fourth SH3 domain which we found is responsible for the interaction to the JNK kinase MKK7. This may explain why POSH and POSH2/SH3RF3 promote, while POSHER inhibits, JNK phosphorylation and activation. In addition, we also showed in this revision that POSH2/SH3RF3 facilitates the binding of JIP3 to MKK7, but does not affect the binding of JIP3 to JNK1 (Fig. 7E and S5E). These new results, together with previous data, indicated that POSH2/SH3RF3 mediates the interaction of MKK7 and JIP3, and then interacts with JNK1 in a JIP3-dependent manner to promote JNK phosphorylation by MKK7.

There are several points that need to improve in order to be published:

- The authors make clear that previous reports firmly established the role of PTX3 in cancer. For this reason, the authors need to check whether it could be a target for cancer treatment. They need to carry on experiments to see if the lack/reduction of PTX3 reduce the tumor size, similarly to their data in Figure 3 targeting POSH2.

Following the suggestion, we analyzed the effect of PTX3 on tumorigenesis. The new data showed that *PTX3* overexpression in HMLER cells enhances tumor incidence in mice (Fig. 4H). Furthermore, *PTX3* knockdown in *POSH2/SH3RF3*-overexpressing MDA-MB-231 cells impaired the POSH2-promoted tumor incidence (Fig. 5H). These data confirmed the role of PTX3 in CSCs and tumor initiation.

- Evaluate the interaction between the component of the signaling network (JIP, JNK, POSH2, MKK)

We performed multiple additional assays to evaluate the interaction among the components of JNK pathway in this revision. First, we performed a sequential IP of POSH2/SH3RF3, MKK7 and JIP3 to prove the presence of these three components in one complex (Fig. 7D). Further, we performed MKK7-JNK and MKK7-JIP3 Co-IP assays in cells with or without *POSH2* overexpression and found that POSH2 facilitates MKK7-JIP3 interaction, while JIP3 binds to JNK in a POSH2-independent manner (Fig. 7E and S5E). Instead, the POSH2-JNK interaction is dependent on JIP3 (Fig. 7F). These data delineated the interaction scenario of these signaling components, indicating that POSH2 mediates the interaction of MKK7 and JIP3, and thus promotes JNK phosphorylation by JIP3-recruited MKK7.

- The use of Ponceau S staining of the nitrocellulose membranes is not acceptable as loading control for the western blots. Antibodies such as tubulin, GAPDH or similar must be employed.

The samples used for PTX3 detection in Western blots (Fig. 4B, D) were conditioned media of cancer cells, to analyze the extracellular level of PTX3. Thus the common loading control of intracellular proteins, such as tubulin and GAPDH was not applicable in this assay. We apologize for that we did not describe the assay clearly and have updated the figure legends in this revision.

- It is important to demonstrate reduced activity of JNK when the interaction between POSH2 and JNK is impaired due to JIP3 knockdown (Fig. 6H). This can be done by showing Jun phosphorylation or the expression of Jun-target genes.

We analyzed JNK activity following *POSH2/SH3RF3* overexpression and *JIP3* knockdown and showed that *POSH2/SH3RF3* enhanced JUN phosphorylation, while *JIP3* knockdown blocked this effect (Fig. 7G).

- Tumor incidence in animal studies should be determined by observing lack of tumors after opening the mice. If this is only determined by external palpation, some very small tumors would not be taken into consideration, and incidence would be underestimated.

We apologize for lack of the accurate description of the animal experiment protocols. In fact, we did surgically open the mammary glands of the mice to confirm tumor incidence at the end point of the experiments for all the *in vivo* assays (Fig. 2E, 3E, 3H, 5G and S1F). We added the description in the Methods section of this revision.

- For the flow cytometry analysis of ALDH⁺ population, the negative control of DEAB⁺ (ALDH inhibitor) should be included, if not, how do the authors really now that the positive population is the real one?

We indeed used DEAB+ as the negative control of all ALDH flow cytometry assays and mentioned it in the Methods section of the previous submission, but we did not show the data as we thought this was a default protocol. We apologize for the negligence and have added in this revision the DEAB+ control data in Fig. 2B, 3B and S3B.

- Authors sort HMLER cells in 2 populations in function of their CD44 expression levels. However, the plots in figure1D show that the level of CD44 in both populations are almost the same. Does CD44LS population increase the levels of CD44 over time?

It is indeed the case. The expression of CD44 gradually increased in different generations of tumorsphere culture from CD44L cells. The 4th generation of passages was named CD44LS and used to analyze ALDH expression in Fig. 1D. The expression of CD44 in various generations of CD44L sphere culture was shown in Fig. S1A of this revision.

- Authors directly relate POSH2 levels with CD44high in HMLE and MCF10AT cell lines and ALDH+ expression in MDA-MB-231. Did the authors measure ALDH+ expression in HMLE and MCF10AT and CD44 in MDA-MB-231?

Previously we had analyzed the contents of ALDH+ cells in HMLE and MCF10AT, as well as CD44⁺CD24⁻ cells in MDA-MB-231. The data showed that HMLE and MCF10AT were mostly ALDH+, while MDA-MB-231 was nearly all CD44⁺CD24⁻ (See Figure_for_review 1 below). This was consistent to the notion that only some specific CSC markers can be used for various cancer cell lines, while other markers are not suitable. For example, it is well known that CD44 is not a good CSC marker for MDA-MB-231. Therefore, we only showed the data with ALDH analysis of MDA-MB-231 and CD44/CD24 analyses of HMLE and MCF10AT in the manuscript.

Figure for review 1.

CD44⁺CD24⁻ percentage in MDA-MB-231 and ALDH⁺ percentage in HMLE and MCF10AT.

- Did the authors perform limiting dilution assays in POSH2 overexpressing-HMLE cell line? Why did the authors specifically use MDA-MB-231 cell line for this experiment? These experiments need to be done also in HMLE cell.

We did not perform the limiting dilution assays with HMLE/HMLER cells in the previous submission because HMLE is an immortalized non-cancerous breast

epithelial cell line without tumor-forming capacity, and its transformed derivative HMLER, although cancerous, has a very low *in vivo* tumorigenic capacity. In addition, HMLER was derived from normal epithelial cells by a series of oncogene transfection and the antibiotic selection markers commonly used for transfection including puromycin, hygromycin and neomycin, were already used during cell line derivation. Therefore, it was difficult to establish *POSH2* stable overexpression line from HMLER.

Nevertheless we have made our efforts to address this concern of the reviewer in this revision. We used blasticidin as the selection marker to establish *POSH2/SH3RF3* overexpression in HMLER cells (Fig. S1E) and injected relatively larger numbers of cells in the limiting dilution assays. The assays showed that *POSH2/SH3RF3* also facilitated *in vivo* tumorigenesis of HMLER cells (Fig. S1F).

- In the siRNA-*POSH2* experiments, did the authors check the siRNA-*POSH2* in the MDA-MB-231 cell line?

We did not perform *POSH2/SH3RF3* knockdown in MDA-MB-231 cells, because its endogenous expression level was very low in MDA-MB-231 (see Figure_for_review 2 below). Therefore, we performed *POSH2/SH3RF3* overexpression in the cell lines with lower *POSH2/SH3RF3* expression (HMLER, MCF10AT and MDA-231) and *POSH2/SH3RF3* knockdown in cell lines with higher *POSH2/SH3RF3* expression, (HMLER-CD44H and MCF10CA1h).

Figure_for_review 2.
POSH2/SH3RF3 expression in various breast cancer cell lines.

- Previously it has been shown that PI3K signalling target pentraxin-3 to promote stemness in basal-like breast cancer (Thomas et al. 2017 Sci Signal. 21;10 (467)). Does *POSH2* exert its functions, at least in part, through the PI3K pathway? Authors should investigate this possibility.

We thank the reviewer for this critical suggestion. We have added the relevant experiments in Fig. S4. It showed that PI3K inhibition could not block the up-regulation of PTX3 expression and tumorsphere formation by *POSH2/SH3RF3* (Fig. S4A and S4B). Moreover, *POSH2/SH3RF3* overexpression did not lead to

PI3K-AKT activation (Fig. S4C). These data demonstrate that POSH2/SH3RF3 regulates PTX3 in a PI3K-independent manner, although it was still possible that PTX3 could also be regulated by PI3K.

- The information about the details in tumorsphere experiment is very deficient. The author should include the cell concentration at the beginning and in the passages.

We have supplied a supplementary table (Table S4) to describe the culturing condition including medium, annexing agents, number of seeded cells and culturing time. We also updated the Methods section in the revision to cite this supplementary table and include more information.

Minor points

- The authors indicate that one band of p-JNK Western blot corresponds to p-JNK1, while the other one corresponds to p-JNK2 (Fig. 6C). However, each band is p-JNK1/2 (two different splice variants of JNK1 and JNK2).

We appreciate the clarification and have corrected our description in this revision.

- In Figures 2 & 3, the tubulin western blot control is clearly overexposed. A shorter exposition (or less protein loaded) must be presented.

We have repeated these western blot assays with loading control over-exposure.

- The authors should show total JNK and Jun protein levels in Fig. 6C.

We followed the suggestion and re-performed these Western blots to include total JNK and JUN protein levels (new Fig. 6E and S4D).

- In the MMTs assays (Figure 1C) authors used paclitaxel amounts between 0-200ng per well, but which volume was used per well? Inhibitors might be indicated by concentration in order to allow reproducibility of the experiments

We apologize in that we meant “ng/mL”, but mistakenly wrote “ng”. We corrected this in the revision. The concentrations of inhibitors were also included in the figure legend.

- The number of replicates is only included in some experiments. Without sample size the presented results don't mean anything. I suggest authors to include this information in their figures.

We added the information of sample sizes and replicate numbers in the figure legends in this version.

Some of the figures are too small and there are some spelling errors.

- “Tumoresphere” should be “tumorsphere” in line of page 5.
- “Overexpression” should be “overexpression” in line 7 of Figure legend 5.
- Line 12 of Methods section lacks the word “in” before “this study”.

We apologize for these errors, and they have been corrected. We also made our efforts to enlarge the figures that were too small in size. We removed the representative tumorsphere image in Fig. 4E which were very small due to space limit and were actually not necessary. We also added one main figure (Fig. 7) and 2 supplementary figures (Fig. S2 and S3) to include the new data without sacrificing the figure sizes. In this submission we also uploaded the original figures in the PPT format.

Reviewer #2 (Remarks to the Author); expert in cancer bioinformatics:

The study Zhang et al present uses a wide array of techniques to investigate the biology of CSCs in breast cancer specifically and cancer more broadly. While most of what is presented in the present manuscript is correct, and conclusions are solid, the manuscript has a severe deficit in some techniques in explaining the methods used. The authors need to be aware that any other scientist that would want to replicate their results need to have the VERY DETAILED and FULL methodology used to produce the presented results, which is not the case. This is reflected in the specific comments below, but not restricted to these:

We appreciate these comments, especially that regarding the methodology description. We have carefully read through the comments and followed the suggestions to include more detailed information.

Throughout all the text:

Minor: Official symbol of the gene is SH3RF3, not POSH2. This needs to be reversed throughout the text (including the title). All other mentioned genes in the manuscript should be also revised to ensure that official gene symbols are used, and not any aliases or old names.

We changed the gene symbol into SH3RF3 through the manuscript and supplementary materials in this new version.

Minor: The authors make several statements like “These results demonstrated”, where “These results suggested” would be more appropriate, since other explanations beyond the scope of this study could explain the changes.

We thank the reviewer for this critical suggestion. We changed the expression “demonstrated” into “suggested” or “indicated” where it was appropriate.

Introduction

Minor: Page 3 line 12: change testified to tested

It was changed as suggested.

Minor: Rephrase the PTX3 sentences, just jumps into it, but needs to start saying first why are they talking about it.

We rephrased the PTX3 sentences as suggested in the new version.

Results

Major: 3 replicas of CD44L (control) and 2 replicas of CD44H and CD44LS. At least 3 replicas of each treatment need to be done.

We believe the comment was referring to Fig. 1E and agree with the reviewer that more replicates would have helped the transcriptomic screening. In fact, we used 3 replicates each of CD44H and CD44L, while one replicate of CD44LS for the transcriptomic analysis. The reason was that our original plan was to look for CSC-associated genes by comparing the transcriptomes between CD44L and CD44H cells and in clinical samples, while CD44LS was only used for a confirmation and backup purpose. Therefore, only one replicate was used for CD44LS though indeed we should have analyzed more replicates. However, now we already finished the screening and identified SH3RF3, and further confirmed its association with cancer stemness in clinical samples and in multiple cell lines including HMLER derivatives (CD44H, CD44L, CD44LS) with qPCR and western analyses. More importantly, functional analyses showed the role of SH3RF3 in CSCs. Thus we think it would be pointless to retrospectively add more replicates for transcriptomic screening at this stage.

Minor: In the mice injected with 40 control cells, were the injection of tumorigenic cells also 40? Furthermore, the increase is “only” 20%, have they controlled in their analysis for the expression of the other already known over or under-expressed cancer genes? Is there a specific gene expression environment in which the overexpression of SH3RF3 is more tumorigenic?

We believe the comment was referring to Fig. 2E and other relevant figures including Fig. 2D. Fig. 2E showed the data of BOTH mouse groups injected with 40 MDA-MB-231 cells of control or *SH3RF3*-overexpressing lines. So the numbers of injected cells were both 40 for control and *SH3RF3*-overexpressing groups. We have updated the figure legend in this revision to avoid the confusion. In fact, Fig. 2E was to show part of the data of the limiting dilution assay in Fig. 2D. In limiting dilution analysis the observation of tumorigenic difference between control and experimental groups only with fewer transplanted cells, rather than more transplanted cells, is a gold standard to indicate CSC regulation. Our data in Fig. 2D and 2E showed that there was no difference in tumorigenesis with 1000 control or *SH3RF3*-overexpressing cells injected per mouse, but the tumor incidence was evidently upregulated in *SH3RF3*-overexpressing group with fewer cells inoculated (200 and 40 cells/mouse). The difference was much larger than 20% (6/10 vs. 1/10 in the 40-cells group, for example) and the *in vitro* analyses also showed evident difference in tumorsphere formation and CSC contents (Fig. 2C and 2B). These results demonstrated the enrichment of stem-like cells by *SH3RF3* overexpression.

In Fig. 2, the comparisons were made between control cells (expressing empty vectors) and *SH3RF3*-overexpressing cells of the same parental cell lines. There was no other gene manipulation in these assays. Thus it may not be necessary or feasible to control other genes, since if there are differences in expression of other genes between the cell groups, the difference should be caused by *SH3RF3* overexpression. Regarding the

gene expression (or genetic environment) for SH3RF3's function in tumorigenesis, it is indeed an important question for cancer studies considering cancer heterogeneity, and that's why we used multiple cell lines for *in vitro* and *in vivo* studies with both *SH3RF3* overexpression and knockdown. The data consistently showed the role of SH3RF3 in CSCs in the tested cell lines, although we could not rule out the possibility that SH3RF3's role is dependent on specific genetic environment. It would be an enormous effort to pin down that environment and we feel it is out of the scope of the current study. Therefore, we updated the Discussion to briefly mention this issue in this revision.

Minor: The 24 genes resulting from the overlap of HMLER and MCF10AT does not imply that they are controlled by SH3RF3, only that they are co-up- or co-down-regulated together with SH3RF3. Not causality, just co-occurrence. As they verify by knockdown, PTX3 seems to be the case, but not necessarily the other 23 genes, rephrase.

Thanks for the suggestion and we have changed the writing in that paragraph to avoid expression such as "regulate", "downstream" in inappropriate places.

Methods

Major: For their own RNAseq data (SH3RF3 and PTX3 over-expressing cell lines), they do not mention absolutely anything about who the data was analysed. They need to state which algorithm was used for reads alignment, how they controlled and/or how they addressed for multimapping of reads, what kind of normalisation they used prior to differential expression analyses, which software/packages they used for all this processed (with references) and which default parameters of these software/packages were changed (if any). Also, which kind of RNAseq was performed? Whole transcriptome or something else? Was the RNAseq paired end or single end? What length of read? How many millions of read per sample they obtained (both raw and after QC)? Which were the QC criteria to remove reads after alignment to reference? Mapping quality? Multimapping? PCR duplicates? Also, were the reads aligned to the hg19 or hg38 of the reference?

We apologize for the negligence and have added a section of RNA sequencing in the Methods to include the detail protocol and relevant information.

Major: Once all this is cleared, they also need to assess how this RNAseq analysis correlates technically with other external analyses considered for the study (e.g. TCGA). Is the RNAseq analysis performed by the authors comparable and compatible with that of the TCGA and others?

Thanks for reminding us of this issue. In this revision, we provided the detail analysis protocol for our internal RNAseq dataset, which was a rather standard protocol performed regularly by a RNAseq service provider (WuXi NextCODE). The same

protocol was used for all the samples of the dataset. In addition, we directly obtained the compiled data matrix of those external datasets (TCGA, GSE7515 and Cancer Cell Line Encyclopedia dataset) for expression comparison analysis. We believe the data processing protocol used for our internal dataset, with proper and previously published algorithms of quality control and normalization, is valid and comparable, although different, to those used for the external datasets. It was difficult to use the same protocol to process our internal dataset as the external datasets given the difference of data formats. Actually, the data processing protocols of these external datasets were also different to each other. Importantly, gene expression comparisons in this study were made only within each internal and external dataset, but not across datasets, and then the overlap of differentially expressed genes of the datasets was identified. Thus, we believe protocol consistency will be necessary only within datasets.

Major: SH3RF3 has genes with similar sequences and implicated in the same processes SH3RF1 and SH3RF2 (as the authors clearly discuss). What is the rate of multimap between the reads that mapped to each of the three genes? How was read multimapping addressed between these 3 genes?

This is an issue which we initially ignored to discuss but could be noteworthy. We performed multimapping analysis for the RNA-seq data, including multimapping of reads of SH3RF1/2/3 to whole transcriptome and cross-mapping among these three genes. As the table below shows, possible multimapping of these reads to other transcripts was all less than 2% and more importantly, there were no cross-mapping of reads among these three genes. We include such analysis results in the Methods of the revised manuscript. We also performed a gene sequence alignment of these 3 homologous genes, and found that there are no regions larger than the length of standard reads with identical sequences in any two of these genes, which might explain the lack of cross-mapping between them.

Statistics of multiple-mapped reads and cross-mapped reads aligned to SH3RF1/2/3

gene	Unique-mapped reads			Multiple-mapped reads		
	SH3RF1	SH3RF2	SH3RF3	SH3RF1	SH3RF2	SH3RF3
sum	2785	6645	94878	56	54	603
rate				2%	0.80%	0.60%

Cross-mapped reads	SH3RF1 vs SH3RF2	SH3RF1 vs SH3RF3	SH3RF2 vs SH3RF3
rate	0	0	0

Reviewer #3 (Remarks to the Author); expert in CSC and breast cancer:

In the submitted manuscript, authors show that POSH2, a scaffold protein increases CSC properties of breast cancer cells through activation of JNK-JUN pathway and Pentaxrin 3. Authors through in vitro tumorsphere formation assay and in vivo experiments coupled with clinical cohort data show the relevance of POSH3 in CSC properties of breast cancer. The manuscript is very well written and communicates the idea convincingly. The role of POSH2 in breast cancer CSC is novel. Given the need to target CSCs to overcome chemo and radio resistant in patients, this study possess clinical relevance.

However, I would like to suggest few corrections before publication of the article:

Major corrections:

1. Throughout the manuscript, authors have used orthotropic mice models to show the role of POSH2 in CSC maintenance. It would interesting to show if POSH2 expression correlates with CSC numbers and function in spontaneous breast cancer models.

We agree that a spontaneous cancer model would be much helpful to further validate the role of POSH2 (in the revise manuscript, we used the official symbol SH3RF3 for this gene following the suggestion of reviewer #2). Since it would be out of the time scope of the current study to construct *POSH2/SH3RF3* knockout mice and cross them into a spontaneous mouse cancer model, we performed the functional analysis of POSH2/SH3RF3 in Py8119 cells, which were derived from the PyMT-driven spontaneous tumors of mice. ALDH⁻ and ALDH⁺ subsets were sorted from the cells by FACS and it was found that the expression of *Sh3rf3* was much higher in ALDH⁺ cells than ALDH⁻ cells (Fig. S1B). Moreover, *Sh3rf3* overexpression in Py8119 enhanced tumorsphere formation (Fig. S1C, D). These results indicated a role of *Posh2* for CSCs in spontaneous tumor-derived breast cancer cells.

2. Authors show that inhibition of PTX3 in control and POSH2 over expressing MDA-MB-231 and HMLER cells inhibit the POSH2 promoted tumorsphere formation. However, no in vivo experiment has been provided to show that inhibiting PTX3 hampers CSC properties of breast cancer cells. It would be good if authors could compare the tumor initiation capability of PTX3 knockdown cells to that of POSH2 knockdown cells.

Following this suggestion, we have added two *in vivo* assays to demonstrate the role of PTX3 in CSCs in this revision, including that of control vs *PTX3*-overexpressing HMLER cells (Fig. 4H) and that of MDA-MB-231 cells with or without *SH3RF3* overexpression and *PTX3* knockdown (Fig. 5H).The data showed that *PTX3* overexpression led to have a stronger tumor-initiating capacity in HMLER cells. In addition, *PTX3* knockdown inhibited tumor initiation and blocked the effect of

SH3RF3 overexpression in MDA-MB-231.

3. In Figure 7, authors have used 3 patient samples, to study the correlation between POSH2 expression and CD44 expression. The sample number is very low to state that there is a positive correlation between the two. Also, in experiments involving patient derived organoids, n=3 which is very less to convincingly demonstrate the positive relationship between POSH2 expression and organoid number.

This might be a confusion caused by imprecise description of sample sizes in the legend of Fig. 7 of the previous version (now Fig. 8 in the revised manuscript). We meant three culturing experiments were performed for Fig. 7A-C by n=3 in the legend, but the sample sizes of patients for the analysis of the correlation of *SH3RF3* expression to CD44 expression or patient survival were much larger than 3 (Fig. 7D-I). In Fig. 7A-C (now Fig. 8A-C), indeed we only used 3 patient-derived breast tumors and 4 patient-derived gastric tumors, but the analyses were not correlative studies. Instead these were functional studies by overexpressing *SH3RF3* in each of the patient-derived tumors and compared the organoid-forming capacity of control and *SH3RF3*-overexpressing tumors. We apologize for this confusion, and have updated the figure legend for better description of the sample sizes.

Minor corrections:

1. In Figure 2A, authors show over expression of POSH2 in HMLE, MCF10AT and MDA-MB-231. Over expression in MCF10AT is minimal while in the other two, tubulin looks saturated. Authors can repeat these blots for better quality. Also, please specify if it is beta tubulin.

We appreciate this comment as it is indeed important for the quality of our data. We repeated the western analyses and optimized the loading amount and exposure time. We also updated the labeling of the loading control in the figure (it was α -TUBULIN that was used in the assay).

2. Figure 2B, upon over expression of POSH2 in HMLE and MCF10AT, authors show increase in CD44⁺CD24⁻ population, while in MDA-MB-231, ALDH⁺ population is increased. What happens to CD44⁺CD24⁻ population in MDA-MB-231?

CD44⁺CD24⁻ and ALDH⁺ are both commonly used CSC markers of breast cancer cell lines; however, not both of them are suitable for all cell lines. It is well known that most of the MDA-MB-231 cells were CD44⁺CD24⁻, which was also confirmed by us (see the figure below); therefore, CD44⁺CD24⁻ was not an appropriate marker to quantitate CSC contents of MDA-MB-231 cells. Similarly, ALDH⁺ might also not be the appropriate CSC marker for MCF10AT and HMLE. That was the reason we used CD44⁺CD24⁻ for MCF10AT and HMLE, while ALDH⁺ for MDA-MB-231.

Figure_for_review 3.
 CD44⁺CD24⁻ percentage in MDA-MB-231 and ALDH⁺ percentage in HMLE and MCF10AT.

3. In Figure 4B, authors show relative expression of PTX3 between control and POSH2 over expression. Control bars are not showing unit “1”, so what is the baseline for “relative expression”?

This was a negligence during figure editing. The y-coordinate should be 2,4,6,8 and the baseline was indeed 1. We corrected it in the new version.

4. Blot for PTX3 in MCF10AT in Figure 4B is not convincing.

We have repeated the analyses by increasing the loaded amount of samples and updated the data in the revision.

5. In Figure 7I, authors show a positive correlation between POSH2 expression and decreased overall survival. Is that relationship true with high PTX3 expression?

We analyzed the expression of PTX3 in the Kaplan-Meier Plotter (KMP) database and did not find a significant correlation of *PTX3* with patient survival, probably due to the fact that the *PTX3* mRNA expression levels of the majority of the KMP samples as analyzed by microarray probe were very low (less than 150). Therefore, we analyzed the protein level of PTX3 in the CPTAC breast cancer cohort and found a correlation of PTX3 with worse prognosis (new Fig. S6 in this revision). However, we could not analyze the expression of SH3RF3 (POSH2) protein in the same cohort as its protein expression level was not available in this dataset. Nevertheless, the correlation of PTX3 expression with prognosis needs further analysis, and we hope to be able to do that in follow-up studies when we have enough specimens with complete prognosis.

Reviewers' comments:

Reviewer #1 (Remarks to the Author):

The authors have satisfactorily responded to all my questions and made the necessary changes to the manuscript.

Reviewer #2 (Remarks to the Author):

With the changes incorporated into the manuscript in answer to the reviewers (this and the others) comments its quality has been greatly improved.

Where I do not provide an answer for a comment it should be assumed that the answer provided by the authors is satisfactory.

We believe the comment was referring to Fig. 1E and agree with the reviewer that more replicates would have helped the transcriptomic screening. In fact, we used 3 replicates each of CD44H and CD44L, while one replicate of CD44LS for the transcriptomic analysis. The reason was that our original plan was to look for CSC-associated genes by comparing the transcriptomes between CD44L and CD44H cells and in clinical samples, while CD44LS was only used for a confirmation and backup purpose. Therefore, only one replicate was used for CD44LS though indeed we should have analyzed more replicates. However, now we already finished the screening and identified SH3RF3, and further confirmed its association with cancer stemness in clinical samples and in multiple cell lines including HMLER derivatives (CD44H, CD44L, CD44LS) with qPCR and western analyses. More importantly, functional analyses showed the role of SH3RF3 in CSCs. Thus we think it would be pointless to retrospectively add more replicates for transcriptomic screening at this stage.

I agree with the authors that this is indeed the case. This reviewer does not argue with the positive finding derived from the RNAseq analysis, validated with qPCR and followed through with functional experiments. What is regrettable is that by not adhering to the standards of RNAseq experiment design and analysis they are most certainly missing out on further relevant results that could have added to the presented results.

We apologize for the negligence and have added a section of RNA sequencing in the Methods to include the detail protocol and relevant information.

With the new included information in the material and methods section now the bioinformatic methodology can be evaluated, and while adequate for the most part some issues need to be addressed:

The authors do not reference how they have dealt with the PCR duplicates.

For the command options specified for STAR, it is understood that they have not performed the two-pass method for STAR alignment which is the best practices pipeline for said software, do the authors have an explanation for why not? This may not be an issue since they obtain very high percentage of aligned reads (mostly thanks to the 150bp long reads as opposed to the more usual 75bp in RNAseq), but any deviation from best practices needs to be explained.

Another best practices point that the authors have neglected is the use of hg38. There are plenty of reasons why the use of hg19 must be discouraged, especially for RNAseq data. Hg19 is already an 11-year-old build of the human genome and its use has been deprecated by all the main best practices pipelines including those of STAR and GATK. Why didn't the authors use hg38?

The authors state that genes with a fold change greater than 2 were identified. Do they mean Log₂(Fold Change) or this is only for the overexpressed genes since they are not doing the Log₂(fold change)?, how did they identify the underexpressed genes in the analysis? Also, what statistical test and multiple test correction/level of significance (e.g. FDR corrected P value < 0.05)

were used?

Reviewer #3 (Remarks to the Author):

Authors have responded to all my comments successfully.

Reviewer #2 (Remarks to the Author):

With the changes incorporated into the manuscript in answer to the reviewers (this and the others) comments its quality has been greatly improved.

Where I do not provide an answer for a comment it should be assumed that the answer provided by the authors is satisfactory.

We believe the comment was referring to Fig. 1E and agree with the reviewer that more replicates would have helped the transcriptomic screening. In fact, we used 3 replicates each of CD44H and CD44L, while one replicate of CD44LS for the transcriptomic analysis. The reason was that our original plan was to look for CSC-associated genes by comparing the transcriptomes between CD44L and CD44H cells and in clinical samples, while CD44LS was only used for a confirmation and backup purpose. Therefore, only one replicate was used for CD44LS though indeed we should have analyzed more replicates. However, now we already finished the screening and identified SH3RF3, and further confirmed its association with cancer stemness in clinical samples and in multiple cell lines including HMLER derivatives (CD44H, CD44L, CD44LS) with qPCR and western analyses. More importantly, functional analyses showed the role of SH3RF3 in CSCs. Thus we think it would be pointless to retrospectively add more replicates for transcriptomic screening at this stage.

I agree with the authors that this is indeed the case. This reviewer does not argue with the positive finding derived from the RNAseq analysis, validated with qPCR and followed through with functional experiments. What is regrettable is that by not adhering to the standards of RNAseq experiment design and analysis they are most certainly missing out on further relevant results that could have added to the presented results.

We completely agree with the reviewer that additional information might be found if more replicates had been used at the beginning. Indeed we plan to perform similar RNA-sequencing analysis, with adequate replicates, on clinical tumors, instead of cancer cell lines, in follow-up studies. We thank the reviewer's suggestion and will take it into consideration in the design of new experiments.

We apologize for the negligence and have added a section of RNA sequencing in the Methods to include the detail protocol and relevant information.

With the new included information in the material and methods section now the bioinformatic methodology can be evaluated, and while adequate for the most part some issues need to be addressed:

The authors do not reference how they have dealt with the PCR duplicates.

We did not remove the PCR duplicates in our RNA sequencing data. PCR duplicates should be taken into consideration in DNA sequencing analyses, in that the genome

has a definite copy number ($n=2$). However, for RNA sequencing, PCR duplicates should be retained in most cases, as duplicate removal might greatly affect the natural duplicates (expression abundance) of RNA species. Only when UMIs (Unique Molecular Identifiers), which was usually for analysis of genes of ultra-low expression, were used in library construction, PCR duplicates could be removed in RNA sequencing. In our study, we had enough RNAs for library construction and only performed a conventional RNA sequencing without UMIs.

We did not state our PCR duplicates retaining in the previous submission as we thought this was a default protocol. We apologize for the negligence and have added the statement in the Method section of this revision.

For the command options specified for STAR, it is understood that they have not performed the two-pass method for STAR alignment which is the best practices pipeline for said software, do the authors have an explanation for why not? This may not be an issue since they obtain very high percentage of aligned reads (mostly thanks to the 150bp long reads as opposed to the more usual 75bp in RNAseq), but any deviation from best practices needs to be explained.

Indeed, two-pass method for STAR alignment is a very useful algorithm. As previous studies indicated (1, 2), it excels on the analyses of RNA alternative splicing, RNA variants and RNA editing. However, in our study, we originally focused only on finding differentially expressed genes in cancer stem-like cells, and thus did not perform two-pass method in the alignment. We thank the reviewer for the comments, and will take the two-pass method into consideration in further studies to identify more relevant information.

Another best practices point that the authors have neglected is the use of hg38. There are plenty of reasons why the use of hg19 must be discouraged, especially for RNAseq data. Hg19 is already an 11-year-old build of the human genome and its use has been deprecated by all the main best practices pipelines including those of STAR and GATK. Why didn't the authors use hg38?

We agree with the reviewer's statement. Regrettably, we started the project in 2015 when hg19 and hg38 were both commonly used. The RNA sequencing were performed at WuXi NextCODE, whose analysis pipelines were mainly based on the hg19 version, at that time. In addition, since our goal of analysis was merely RNA expression difference, it was performed with the regular pipeline.

Again we appreciate the reviewer's rigor on bioinformatic protocols and admit that as a research group that does not primarily focused on bioinformatics, we used an analysis pipeline that might not be optimized in certain steps, and this might have possibly led to the missing of some additional information. Certainly we will take note of this in future studies.

The authors state that genes with a fold change greater than 2 were identified. Do they mean $\text{Log}_2(\text{Fold Change})$ or this is only for the overexpressed genes since they are not doing the $\text{Log}_2(\text{fold change})$?, how did they identify the underexpressed genes in the analysis? Also, what statistical test and multiple test correction/level of significance (e.g. FDR corrected P value < 0.05) were used?

We apologize for the unclear description. We meant both the up-regulated and down-regulated genes with fold changes > 2 (case/control > 2 for up-regulated genes and < 0.5 for down-regulated genes). The R package EdgeR was used for differential expression analysis and the statistical test used in EdgeR was exact test, with Benjamini-Hochberg FDR correction of the p values. We have updated the description and added necessary information in this version.

References:

1. Wilson, G.W., and Stein, L.D. 2015. RNASequel: accurate and repeat tolerant realignment of RNA-seq reads. *Nucleic Acids Res* 43:e122.
2. Veeneman, B.A., Shukla, S., Dhanasekaran, S.M., Chinnaiyan, A.M., and Nesvizhskii, A.I. 2016. Two-pass alignment improves novel splice junction quantification. *Bioinformatics* 32:43-49.

REVIEWERS' COMMENTS:

Reviewer #2 (Remarks to the Author):

I commend the authors for their thorough and knowledgeable answers to my queries. I accept all answers as satisfactory but one:

 We did not remove the PCR duplicates in our RNA sequencing data. PCR duplicates should be taken into consideration in DNA sequencing analyses, in that the genome has a definite copy number (n=2). However, for RNA sequencing, PCR duplicates should be retained in most cases, as duplicate removal might greatly affect the natural duplicates (expression abundance) of RNA species. Only when UMIs (Unique Molecular Identifiers), which was usually for analysis of genes of ultra-low expression, were used in library construction, PCR duplicates could be removed in RNA sequencing. In our study, we had enough RNAs for library construction and only performed a conventional RNA sequencing without UMIs.

We did not state our PCR duplicates retaining in the previous submission as we thought this was a default protocol. We apologize for the negligence and have added the statement in the Method section of this revision.

The matter of whether PCR duplicates need to be removed or not in RNAseq studies is an open one, with partial evidence leaning either side depending on the situation. I agree with the authors in that in the case of high quality, high amount starting material, removal of PCR duplicates can be neutral or even harmful in the case of very short reads (50bp or below) which is not the case of this study's data. Even in the cases (which given the author's comments would be this case) in which removal of PCR duplicates is not necessary, evidence for this is required in the form of the actual average percentage of PCR (and standard deviation) for reviewers and readers evaluation that this is the case indeed. To be clear: the fact that the percentage is stated does not mean that they are actually removed from analysis. A sentence in the M&M with average PCR duplicates and deviation should suffice.

Reviewer #2 (Remarks to the Author):

The matter of whether PCR duplicates need to be removed or not in RNAseq studies is an open one, with partial evidence leaning either side depending on the situation. I agree with the authors in that in the case of high quality, high amount starting material, removal of PCR duplicates can be neutral or even harmful in the case of very short reads (50bp or below) which is not the case of this study's data. Even in the cases (which given the author's comments would be this case) in which removal of PCR duplicates is not necessary, evidence for this is required in the form of the actual average percentage of PCR (and standard deviation) for reviewers and readers evaluation that this is the case indeed. To be clear: the fact that the percentage is stated does not mean that they are actually removed from analysis. A sentence in the M&M with average PCR duplicates and deviation should suffice.

Thank you for your advice. We have learned a lot in this reviewing process. We estimated the PCR duplicate rates and added the relevant statement in the Methods section.